# BLISS: A Lightweight Bilevel Influence Scoring Method for Data Selection in Language Model Pretraining

## Abstract

Effective data selection is essential for pretraining large language models (LLMs), enhancing efficiency and improving generalization to downstream tasks. However, existing approaches often require leveraging external pretrained models, making it difficult to disentangle the effects of data selection from those of the external pretrained models. In addition, they often overlook the long-term impact of selected data if the model is trained for a long period of time, primarily due to the prohibitive cost of full-scale LLM pretraining. In this paper, we introduce BLISS (**B**ileve**L** **I**nfluence **S**coring method for data **S**election): a lightweight data selection method that operates entirely *from scratch*, without relying on any external pretrained oracle models, while explicitly accounting for the long-term impact of selected data. BLISS leverages a small proxy model as a surrogate for the LLM and employs a score model to estimate the long-term influence of training samples if the proxy model is trained to convergence. We formulate data selection as a bilevel optimization problem, where the upper-level objective optimizes the score model to assign importance weights to training samples, ensuring that minimizing the lower-level objective (i.e., training the proxy model over the weighted training loss until convergence) leads to best validation performance. Once optimized, the trained score model predicts influence scores for the dataset, enabling efficient selection of high-quality samples for LLM pretraining. We validate BLISS by pretraining 410M/1B/2.8B Pythia and LLaMA-0.5B models on selected subsets of the C4 dataset. Notably, under the 1B model setting, BLISS achieves $1.7\times$ speedup in reaching the same performance as the state-of-the-art method, demonstrating superior performance across multiple downstream tasks.

## 1 Introduction

The successful large-scale language model pretraining crucially relies on the careful choice of pretraining data (Brown et al., 2020; Raffel et al., 2020; Du et al., 2022; Elazar et al., 2023). Recent studies have shown that effective data selection (a.k.a., data curation) methods can enhance pretraining efficiency (Xie et al., 2023a) and improve generalization (Engstrom et al., 2024; Wettig et al., 2024). There are various types of data selection approaches for language model pretraining, including language filtering (Laurençon et al., 2022; Wenzek et al., 2019), data deduplication (Lee et al., 2021; Abbas et al., 2023), heuristic approaches (Rae et al., 2021; Penedo et al., 2023), data quality data filtering (Brown et al., 2020; Gao et al., 2020; Chowdhery et al., 2023; Xie et al., 2023b; Wettig et al., 2024), data mixing (Xie et al., 2023a; Albalak et al., 2023; Xia et al., 2023), and data influence function based methods (Park et al., 2023; Engstrom et al., 2024; Yu et al., 2024). Despite the rich literature of data selection methods in large language model (LLM) pretraining (e.g., a survey paper in Albalak et al. (2024)), it is still unclear what properties are needed for the training data curation to guarantee good performance: it remains an important real-world challenge (Li et al., 2024).

Existing approaches of data selection methods suffer from two major limitations. First, they often require leveraging pretrained models (Brown et al., 2020; Xie et al., 2023b; Wettig et al., 2024) for data-quality filtering, making it difficult to separate the effects of data selection from those of the external pretrained models. For example, the QuRating method (Wettig et al., 2024) assigns quality ratings to training samples based on responses from a pretrained LLM (e.g., GPT-3.5) before training a QuRater model. This reliance raises uncertainty about the role of the external LLM in the training process and whether its feedback is truly optimal. Moreover, the cost of invoking these external pretrained models is prohibitively expensive during data selection process for large-scale pretraining.

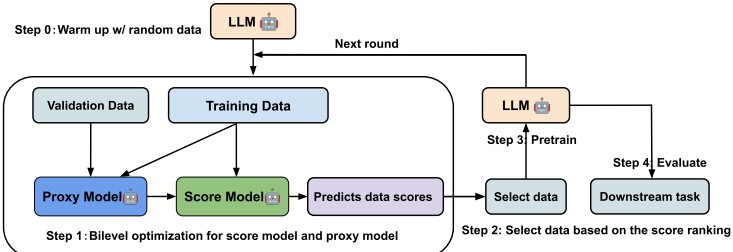

Figure 1: The pipeline of data selection and pretraining procedure. There are four main steps in one round training, 1) Warm up LLM using randomly selected training data (e.g. 10k step); 2) Bilevel optimization for score and proxy model, 3) Predict the data influence, and select Top-20% training data based on their score ranking; 4) Retrain the LLM using the selected data (e.g., 10k steps); 5) Evaluate on the downstream task. Repeating the above steps can achieve multiple-round training.

Second, they typically do not consider the long-term impact of selected data if the model is trained for a long time (i.e., multiple steps of gradient-based updates). For example, the data influence function based approach (Yu et al., 2024) evaluates the impact of individual training samples based on a single training step with the current model, which does not capture the cumulative effects of data selection over the course of full model training.

In this paper, we introduce a new data selection method, to address the two major limitations of existing approaches. Our method, namely BLISS (**B**ileve**L** **I**nfluence **S**coring method for data **S**election), is a lightweight data selection method that operates entirely *from scratch*, without relying on any external pretrained models, while explicitly accounting for the long-term impact of selected data. The core innovation of our approach lies in *the integration of two lightweight models within a novel bilevel optimization framework* for data selection. Our method bypasses traditional data-quality filtering and explicitly considers the long-term impact of selected data throughout training. In particular, BLISS leverages a small proxy model as a surrogate for the LLM and employs a score model to estimate the long-term influence of training samples if the proxy model is trained to convergence. Our bilevel optimization problem has upper-level and lower-level objectives: the upper-level objective optimizes the score model to assign importance weights to training samples, ensuring that minimizing the lower-level objective (i.e., training the proxy model over the weighted training loss until convergence) leads to best validation performance. Once the bilevel optimization is solved, the trained score model predicts influence scores for the entire dataset, enabling the selection of high-score samples for LLM pretraining. The pipeline of our proposed procedure is illustrated in Figure 1. The main contributions of our paper are summarized as the following:

- We propose a principled approach to data selection for language model pretraining. Our method, BLISS, leverages a novel bilevel optimization framework that employs a proxy model and a score model to explicitly account for the long-term impact of selected data. Unlike existing methods, BLISS operates from scratch without relying on any pretrained oracle models for data-quality filtering, obviating any biases or risks that may arise from such dependence[1].

- We validate our method by pretraining 410M/1B Pythia and LLaMA-0.5B models on selected subsets of C4 dataset. Experimental results on 1B setting demonstrate a $1.7\times$ speedup in reaching the same performance as the state-of-the-art method such as MATES (Yu et al., 2024). Furthermore, we scale up our experiments to a 2.8B model pretraining used by the data selected in the 1B experiment, and we demonstrate that our method consistently outperforms MATES at every round of data selection, achieving $1.4\%$ performance improvement over MATES (Yu et al., 2024).

- Through extensive ablation studies, we demonstrate the effectiveness of each component in our bilevel optimization framework, further substantiating the robustness and efficiency of our approach.

---

[1]Many commercial large-scale pretrained models strictly prohibit users from generating data or using them to facilitate the training of other models, as doing so may result in severe legal consequences (OpenAI, 2024; Google, 2024). Our approach is entirely free from such legal concerns. We rely solely on algorithmic advancements applied to a model trained from scratch, without any dependence on third-party pretrained large-scale models.

## 2 RELATED WORK

**Data Selection for Language Model Training.** Early approaches to data selection primarily relied on rule-based methods as language filters for training data, employing utility functions tailored to specific datasets (Conneau & Lample, 2019; Raffel et al., 2020; Rae et al., 2021; Penedo et al., 2023). Another key category is data deduplication (Lee et al., 2021; Sorscher et al., 2022; Penedo et al., 2023; Abbas et al., 2023; Tirumala et al., 2023), which eliminates redundant samples to optimize training efficiency and enhance performance on downstream tasks. A class of methods exist for performing data-quality filtering, which can select data similar to high-quality corpus of data points (Brown et al., 2020; Du et al., 2022; Gao et al., 2020; Xie et al., 2023b; Li et al., 2024), with small perplexity (Chowdhery et al., 2023; Wenzek et al., 2019). More recent methods leverage external pretrained LLMs to evaluate the pretraining data quality (Wettig et al., 2024; Maini et al., 2024; Zhuang et al., 2025). In addition, a similar variant of data selection is domain reweighting for data mixtures (Oren et al., 2019; Sagawa et al., 2019; Xie et al., 2023a; Fan et al., 2023; Albalak et al., 2023; Chen et al., 2024), which re-scale the contribution of each domain to enhance generalization. Another recently emerged line of research leverages the tool of influence functions (Hampel, 1974; Cook, 1977; Ling, 1984; Koh & Liang, 2017) to evaluate the impact of individual training samples on a fixed LLM (Park et al., 2023; Engstrom et al., 2024; Yu et al., 2024; Pan et al., 2025; Lin et al., 2024). QUAD (Zhang et al., 2024) proposes an efficient framework incorporating the attention layers to estimate the influence scores.

In contrast to these works, our work explicitly considers the long-term impact of selected data if the model is not simply fixed but trained for a long time. In addition, our method can train the model from scratch and does not need any extra information from any external pretrained models, making it a scalable and effective solution.

**Bilevel Optimization and Data Selection.** Bilevel optimization provides a powerful framework for modeling optimization problems with a nested structure (Bracken & McGill, 1973; Dempe, 2002). Recent research has focused on developing efficient bilevel optimization algorithms with strong theoretical guarantees (Ghadimi & Wang, 2018; Hong et al., 2023; Ji et al., 2021; Kwon et al., 2023; Dagréou et al., 2022; Chen et al., 2023; Grazzi et al., 2022; Hao et al., 2024; Gong et al., 2024). This approach has been widely applied in various machine learning tasks, including meta-learning (Finn et al., 2017), hyperparameter optimization (Franceschi et al., 2018), and natural language processing (Somayajula et al., 2023; Grangier et al., 2023). For the application of data selection, bilevel optimization has been utilized for continual learning (Borsos et al., 2020; Zhou et al., 2022; Hao et al., 2023) and data reweighting in LLM fine-tuning (Pan et al., 2024; Shen et al., 2024). Our work is most closely related to SEAL (Shen et al., 2024), which focuses on selecting high-quality and safe data to fine-tune a pretrained LLM, with the goal of aligning the model with safety and ethical guidelines. However, our approach differs from SEAL in two key aspects: (1) Problem setting. While SEAL operates in a fine-tuning context, our objective is to select data for **pretraining** an LLM **from scratch**, aiming to improve downstream performance **without relying on any external pretrained models**. (2) Model update mechanism. SEAL utilizes the LoRA technique (Hu et al., 2021) to update both the data selector and the LLM during fine-tuning. However, this approach is not directly applicable to our setting due to the following reasons. First, LoRA is only suitable for fine-tuning tasks but insufficient for full model pretraining. Second, their algorithm always updates the original large models directly, which is computationally expensive if all parameters are updated. In contrast, we propose a more efficient framework that introduces lightweight models (a score model and a proxy model) to guide data selection, while allowing full parameter updates within these smaller networks. To the best of our knowledge, our proposed bilevel influence scoring method is the first to leverage bilevel optimization techniques for data selection in LLM pretraining.

## 3 PRELIMINARIES AND NOTATIONS

Suppose that we have a large-scale training dataset $\mathcal{D}_{tr} = \{\xi_i \mid 0 \leq i \leq N-1\}$ and a downstream task $\mathcal{D}_{ds}$. The goal is to select a subset of training set, namely $\mathcal{D}_s = \{\xi_j \mid 0 \leq j \leq Q-1, Q \leq N\}$, to pretrain a large language model with a specific training budget (e.g., limited FLOPs), such that the model can achieve high performance on the downstream task $\mathcal{D}_{ds}$. Generally, the downstream data is inaccessible during pretraining. Instead, we can use a validation data $\mathcal{D}_{val} = \{\zeta_i \mid 0 \leq i \leq M-1\}$ to estimate the model's performance on $\mathcal{D}_{ds}$, because these two datasets often have similar data distributions or share common domain knowledge. A small subset of training data $\tilde{\mathcal{D}}_{tr} \subset \mathcal{D}_{tr}$ is uniformly sampled from $\mathcal{D}_{tr}$.

In bilevel optimization, $f(\cdot)$ and $g(\cdot)$ denote the upper-level (UL) and lower-level (LL) functions, respectively. Machine learning often requires solving stochastic optimization problems $f(\cdot) = \mathbb{E}_{\xi \sim \mathcal{D}_f}[F(\cdot; \xi)]$ and $g(\cdot) = \mathbb{E}_{\zeta \sim \mathcal{D}_g}[G(\cdot; \zeta)]$, where $\mathcal{D}_f$ and $\mathcal{D}_g$ are the underlying unknown data distribution for $f$ and $g$, respectively. $F(\cdot; \xi)$ denotes the upper-level stochastic objective function and $G(\cdot; \zeta)$ is the lower-level stochastic objective function. Noisy observations of $f$ and $g$ can be collected by sampling from $\mathcal{D}_f$ and $\mathcal{D}_g$.

# 4 METHODS

## 4.1 BILEVEL INFLUENCE SCORING FRAMEWORK

The goal of data selection is to optimize the performance of the LLM on downstream tasks by training it using an optimal subset of training data. However, directly searching for the optimal subset of training samples faces prohibitive costs due to the combinatorial nature of the problem and the high computational cost of estimating the performance of the LLM for every potential subset being evaluated.

To address the aforementioned computational challenge, our bilevel influence scoring framework uses a lightweight score model $\theta_s$ to predict the influence of every sample on the model's performance for the downstream task. The optimized score model is then used to infer the influence score of training samples, enabling the selection of the subset with the highest influence, thus streamlining the process to search for the optimal training data. Instead of directly estimating the performance of LLM (parameterized by $\theta_{tr}$) which is computationally expensive, our framework introduces a lightweight proxy model $\theta_p$ to approximate the behavior of the LLM. Note that the score model and the proxy model are both small models: they share a similar architecture and number of parameters. To ensure the data preferences of the proxy model align with those of the LLM, we apply knowledge distillation by minimizing the Kullback-Leibler (KL) divergence between the output logits of the proxy model and the LLM. We formulate the bilevel optimization for data selection as follows:

$$\min_{\theta_s} \Phi(\theta_s) \coloneqq f(\theta_p^*(\theta_s)) \coloneqq \mathbb{E}_{\zeta \sim \mathcal{D}_{\text{val}}} F(\theta_p^*(\theta_s); \zeta) \quad \text{(UL)},$$

$$\text{s.t.} \quad \theta_p^*(\theta_s) \in \arg\min_{\theta_p} g(\theta_p, \theta_s) \coloneqq \mathbb{E}_{\xi \sim \mathcal{D}_{tr}} G(\theta_p, \theta_s; \xi) \quad \text{(LL)}. \tag{1}$$

where $\mathbb{E}_{\xi \sim \mathcal{D}_{tr}} G(\theta_p, \theta_s; \xi) = \sum_{i=0}^{N-1} P_i \mathcal{L}(\theta_p; \xi_i) + \gamma D_{KL}\left(\ell(\theta_p; \xi_i) \| \ell(\theta_{tr}; \xi_i)\right) + \lambda \|\theta_p\|^2$ and $P_i = \frac{e^{h(\theta_s; \xi_i)}}{\sum_{j=1}^N e^{h(\theta_s; \xi_j)}}$ represents the importance weight of sample $i$, and $h(\cdot) : \mathbb{R}^{d_x} \to (0, 1)$ is a function that maps a sample from $\mathbb{R}^{d_x}$ to an influence score in the range $(0, 1)$. $\mathcal{L}(\cdot)$ and $F(\cdot)$ denote the loss functions for next token prediction, with a common choice being cross-entropy. The model's output logits are represented by $\ell(\cdot)$. The KL divergence is defined as $D_{KL}(X \| Y) = \sum_i X_i \log(\frac{X_i}{Y_i})$. $\gamma$ and $\lambda$ are the regularization coefficients for the KL divergence and the weight decay terms, respectively.

BLISS evaluates the long-term influence of training samples if the proxy model is trained to its convergence state $\theta_p^*(\theta_s)$. Specifically, the lower-level trains the proxy model on the weighted training loss until convergence. This is notably different from other methods such as MATES (Yu et al., 2024), which trains a single step on the selected data from the current model state before evaluating sample influence. Consequently, MATES overlooks the long-term influence of training samples and may not fully capture the importance of data for downstream tasks. It is also worth noting that the bilevel data selection framework, described in formula (1), does not rely on any external pretrained models which are typically trained on large-scale natural language corpora. This independence makes BLISS a more self-contained approach to data selection, which also obviates any biases or risks associated with external pretrained models that may involve proprietary or sensitive data.

## 4.2 ALGORITHM FOR UPDATING THE PROXY MODEL AND SCORE MODEL

Now we design efficient algorithms for solving the bilevel problem (1). The lower-level problem aims to optimize the proxy model $\theta_p$ on the weighted training samples with the influence predicted by the score model. Note that we freeze the LLM ($\theta_{tr}$) through the process of solving the bilevel optimization problem, as the LLM is used to infer the output logits. Therefore, we perform the

following update for the lower-level objective on a mini-batch of size $\mathcal{B}$:

$$\theta_p^{t+1} = \theta_p^t - \eta_1 \nabla_{\theta_p} \sum_{i=1}^{\mathcal{B}} G(\theta_p^t, \theta_s^t; \xi_i)$$

$$= \theta_p^t - \eta_1 \sum_{i=1}^{\mathcal{B}} \left( P_i \nabla_{\theta_p} \mathcal{L}(\theta_p^t; \xi_i) + \gamma \sum_j \nabla_{\theta_p} \ell_j(\theta_p^t; \xi_i) \log \frac{\ell_j(\theta_p^t; \xi_i)}{\ell_j(\theta_{tr}^t; \xi_i)} + 2\lambda \theta_p^t \right), \tag{2}$$

where $\ell_j(\cdot)$ denotes the $j$-th logit of the output. Note that the exact computation of $P_i$ depends on all $N$ samples, which is computationally infeasible. Therefore, we approximate $P_i$ by replacing the full summation in the denominator with a partial summation over a smaller subset. This approximation is implemented in a distributed manner, significantly reducing the computational overhead. More details can be found in Appendix G. For the upper-level update, we take the derivative of $\Phi(\theta_s)$ with respect to $\theta_s$ by chain rule, which is known as the hypergradient:

$$\nabla_{\theta_s} \Phi(\theta_s) = -\nabla_{\theta_s \theta_p}^2 g(\theta_p^*(\theta_s), \theta_s) \underbrace{[\nabla_{\theta_p}^2 g(\theta_p^*(\theta_s), \theta_s)]^{-1} \nabla_{\theta_p} f(\theta_p^*(\theta_s))}_{z}, \tag{3}$$

where $z$ is the solution of the quadratic function $\min_z \frac{1}{2} z^T \nabla_{\theta_p}^2 g(\theta_p^*(\theta_s), \theta_s) z - z^T \nabla_{\theta_p} f(\theta_p^*(\theta_s))$. It can be solved by running a few steps of gradient descent in practice:

$$z_{k+1}^t = z_k^t - \eta_2 \left( \nabla_{\theta_p}^2 g(\theta_p^t, \theta_s^t) z_k^t - \nabla_{\theta_p} f(\theta_p^t) \right), \tag{4}$$

where $k$ is the number of gradient updates for updating $z$ at a fixed iteration $t$ of updating $\theta_s$. We run 3 steps of gradient descent to solve $z$ in our experiments. Note that Equation (4) computes the Hessian-Vector-Product (HVP) term $\nabla_{\theta_p}^2 g(\theta_p^t, \theta_s^t) z_k^t$ and thus avoids the computationally prohibitive operation of taking the inverse of the Hessian. The dimension of $z$ is the same as that of the parameters of the lightweight proxy model. Therefore, the computation of HVP within the PyTorch framework is quite similar to that of gradient. In our implementation, we use the stochastic variants of Equation (3) and Equation (4) for updating the score model. In particular, the approximation of hypergradient at iteration $t$ on the mini-batch $\mathcal{B}$ is

$$\nabla_{\theta_s} \widehat{\Phi}(\theta_s^t) = -\sum_{i=1}^{\mathcal{B}} P_i \nabla_{\theta_s} h(\theta_s^t; \xi_i) \nabla_{\theta_p} \mathcal{L}(\theta_p^t; \xi_i) z^t + \sum_{i=1}^{\mathcal{B}} P_i \sum_{j=1}^{\mathcal{B}} P_j \nabla_{\theta_s} h(\theta_s^t; \xi_j) \nabla_{\theta_p} \mathcal{L}(\theta_p^t; \xi_i) z^t. \tag{5}$$

Then the update for the upper-level variable ($\theta_s$) is $\theta_s^{t+1} = \theta_s^t - \eta_3 \nabla_{\theta_s} \widehat{\Phi}(\theta_s^t)$. When the score model converges over $T$ steps, reaching $\theta_s^T$, it is then used to estimate the influence scores of the entire training dataset in the current round by: $S_i = h(\theta_s^T, \xi_i), \forall \xi_i \in \mathcal{D}_{tr}$. Then the influence scores are collected: $\{S_i \mid 0 \le i \le |D_{tr}|\}$, and the top-ranked samples with the highest influence scores are selected to construct $\mathcal{D}_s$, which is used to pretraining the LLM ($\theta_{tr}$).

The detailed implementation of the algorithm is presented in Algorithm 1. In practice, we use `Adam` optimizer (Kingma & Ba, 2014) to update the upper-level variables, where we will update the Adam gradient with the calculated hypergradient. The pretraining process is conducted over $R$ rounds. In each round, the algorithm performs data selection followed by LLM retraining. The training dataset is partitioned into $R$ shards. The data selection in round $r$ is conducted on $\mathcal{D}_{tr}^r$. The LLM resumes training from the previous round's checkpoint and updates to $\theta_{tr}^r$ at the end of the $r$-th round. Similarly, the score model also continues learning throughout the process, reaching $\theta_s^r$ at the $r$-th round. It is worth noting that the proxy model ($\theta_p^r$) is reinitialized with the warm-up model at the beginning of each round. This prevents the model from overfitting to the previous round's training data and ensures it can better capture the evolving behavior of the LLM.

### 4.3 WARM UP MODELS

The key distinction between our algorithm and other data selection methods (Brown et al., 2020; Xie et al., 2023b; Wettig et al., 2024) is that it operates independently of external pretrained models, avoiding biases from data selection influenced by such models. However, without leveraging pretrained knowledge, the proxy model, score model, and LLM tend to perform poorly in the initial phase due to random parameter initialization. To mitigate this issue, we incorporate a model warm-up step before data selection, similar to other data selection approaches (Yu et al., 2024; Xia et al., 2024), using

---

**Algorithm 1** `BLISS`

---

1: **Input:** $\eta_1, \eta_2, \eta_3, R, T, K, Q, \mathcal{D}_{tr}, \tilde{\mathcal{D}}_{tr}, \mathcal{D}_{val}$
2: **Initialize:** Warm up $\theta_p^{0,0}, \theta_s^{0,0}, \theta_{tr}^{0,0}$ using randomly selected training data.
3: **for** $r = 0, \ldots, R - 1$ **do**
4:     $\theta_p^{0,r} = \theta_p^{0,0}$                                   # reset proxy/score parameters for the new round
5:     $\theta_s^{0,r} = \theta_s^{T,r-1}$ if $r > 1$ else $\theta_s^{0,0}$
6:     $\theta_{tr}^{0,r} = \theta_{tr}^{Q,r-1}$ if $r > 1$ else $\theta_{tr}^{0,0}$
7:     **for** $t = 0, \ldots, T - 1$ **do**
8:         Sample $\xi_t^r, \tilde{\xi}_t^r, \pi_t^r \leftarrow \tilde{\mathcal{D}}_{tr}^r$, and sample $\zeta_t \leftarrow \mathcal{D}_{val}$
9:         $\theta_p^{t+1,r} = \theta_p^{t,r} - \eta_1 \nabla_{\theta_p} G(\theta_p^{t,r}, \theta_s^{t,r}; \xi_t^r)$          # LL: update the proxy model for lower-level
10:        $z^{t+1,r} = \text{GDLS}(\eta_2, K, \nabla_{\theta_p} G(\theta_p^{t,r}, \theta_s^{t,r}; \tilde{\xi}_t^r), \nabla_{\theta_p} F(\theta_p^{t,r}, \theta_s^{t,r}; \zeta_t))$     # solve the linear system

11:        $\theta_s^{t+1,r} = \text{Adam}(\theta_s^{t,r}, -\nabla_{\theta_s \theta_p}^2 G(\theta_p^{t+1,r}, \theta_s^{t,r}; \pi_t^r) z^{t+1,r}, \eta_3)$ [2]   # UL: update the score model
12:     **end for**
13:     Infer the influence score $\{S_i^r \mid 0 \leq i \leq |\mathcal{D}_{tr}^r| - 1\}$ on $\mathcal{D}_{tr}^r$ using $\theta_s^{T,r}$
14:     Sort $\{S_i^r\}$ in descending order and select the 20% data with the highest influence scores from $\mathcal{D}_{tr}^r$ to form the selected data $\mathcal{D}_s$
15:     **for** $\tau = 0, \ldots, Q - 1$ **do**
16:         Sample $\xi_\tau$ from $\mathcal{D}_s$.
17:         $\theta_{tr}^{\tau+1,r} = \theta_{tr}^{\tau,r} - \eta_4 \nabla_{\theta_{tr}} \ell(\theta_{tr}^{\tau,r}; \xi_\tau)$ # pretrain the LLM
18:     **end for**
19: **end for**

---

**Algorithm 2** `GDLS: Gradient Descent for the Linear System Solution`

---

1: **Input:** $\eta, K, \nabla_{\theta_p} g(\theta_p), a$
2: **Initialize:** $z_0$
3: **for** $k = 0, \ldots, K - 1$ **do**
4:     $z_k = z_{k-1}$ if $k > 1$ else $z_0$
5:     $z_{k+1} = z_k - \eta\left(\nabla_{\theta_p}^2 g(\theta_p) z_k - a\right)$
6: **end for**
7: Return $z_K$

---

randomly selected samples. The lightweight proxy and score models share token embedding layers and transformer blocks but differ in their final layers: the proxy model handles token generation, while the score model outputs influence scores for individual samples. Consequently, only the proxy model and the LLM require warm-up, while the score model can be initialized with the weights from proxy model directly.

## 5 EXPERIMENTS

In this section, we validate the proposed bilevel influence scoring framework for pretraining data selection. We apply the bilevel optimization algorithm to train a lightweight proxy model ($\theta_p$) and a score model ($\theta_s$)) for data selection. We then pretrain a target LLM ($\theta_{tr}$), specifically Pythia-410M/1B, from scratch on a selected subset of the large-scale C4 dataset (Raffel et al., 2020), which is designed for LLM pretraining. we then evaluate the pretrained LLM on multiple downstream tasks and compare its performance against several baseline methods, including Random selection, DSIR (Xie et al., 2023b), SemDeDup (Abbas et al., 2023), DsDm (Engstrom et al., 2024), LESS (Xia et al., 2024), QuRating (Wettig et al., 2024), and MATES (Yu et al., 2024). We furthermore scale up our experiment to 2.8B model pretraining and achieve $1.4\%$ performance improvement over the state-of-the-art method. The domain reweighting experiment is deferred to Appendix J.

### 5.1 DATASET SETTINGS

Following the approach of DsDm (Engstrom et al., 2024), we perform data selection and pretraining using tokenized data. The procedure of BLISS is implemented for 5 rounds (i.e., $R = 5$), with the

---

[2] `Adam(variable, gradient, lr)` optimizer receives the current variable, its hypergradient and learning rate. Then it updates the first and second momentum, then returns the updated variable.

C4 dataset partitioned into five equal shards, denoted as $\{\mathcal{D}_{tr}^r \mid 0 \leq r \leq 4\}$. Each training round operates on a distinct data shard without replacement. In every round, we first uniformly sample a small proportion (0.1%) from $\mathcal{D}_{tr}^r$ as the bilevel training set $\tilde{\mathcal{D}}_{tr}^r$ for updating the proxy model. We use LAMBADA (Paperno et al., 2016) as validation data for updating the score model. Other datasets, including ARC-E (Clark et al., 2018), SQUAD (Rajpurkar, 2016), and PIQA (Bisk et al., 2020), are evaluated in the ablation study (Appendix C.4).

To evaluate the performance of data selection algorithms, we run the pretraining model across 9 downstream tasks, including SciQ (Welbl et al., 2017), ARC-E (Clark et al., 2018), ARC-C (Clark et al., 2018), LogiaQA (Liu et al., 2020), OBQA (Mihaylov et al., 2018), BoolQ (Clark et al., 2019), HellaSwag (Zellers et al., 2019), PIQA (Bisk et al., 2020), and WinoGrande (Sakaguchi et al., 2021). These tasks cover a diverse range of reasoning and comprehension challenges, including question answering, logical inference, commonsense reasoning, and coreference resolution. Thus it requires models to demonstrate various capabilities, such as retrieving and applying scientific knowledge, understanding causal relationships, resolving ambiguities in natural language, and making informed choices among distractors. A good data selection algorithm is expected to select the "important" data that boost model performance across these downstream tasks.

## 5.2 Model Settings

The target pretraining model, Pythia-410M/1B/2.8B, consists of 410 million, 1 billion or 2.5 billion trainable parameters. Both the proxy model and score model are based on Pythia-31M (for Pythia-410M) or Pythia-160M (for Pythia-1B), but they serve different purposes: the proxy model acts as a surrogate for the LLM and is trained for next-token prediction, while the score model functions as a regression model that maps individual samples to corresponding influence scores. Details of model settings are deferred to Appendix A. Notably, all models are trained from scratch using Gaussian initialization for model parameters. Additional experimental details, including hyperparameter choices, learning rate schedules, and distributed training strategies, are provided in Appendix E.

## 5.3 Bilevel Optimization for Proxy Model and Score Model

In the Pythia-410M setting, the proxy model $\theta_p$ is updated with a "single-step" optimization per iteration (line 9 in Algorithm 1). However, when scaling up to larger models like Pythia-1B, we adopt a "multi-steps" update strategy for the proxy model to achieve a better lower-level solution. To demonstrate the effectiveness of bilevel optimization in training the proxy model and score model, we visulize the evolution of the training loss at during round 2 (Figure 6(a) in Appendix F) and round 5 (Figure 6(b) in Appendix F). Since the first round uses randomly selected data to warm up the LLM, our data selection algorithm is employed from the second round onward.

Within each round, both losses exhibit a two-phase trend: they initially decrease rapidly before experiencing a slight increase. This behavior arises due to the composition of the lower-level objective function, which includes three terms: the weighted cross-entropy loss, the KL divergence loss, and a regularization term. In the first phase, the weighted cross-entropy loss dominates, decreasing as the proxy model is optimized. In the second phase, the KL divergence term becomes more influential. Since the LLM has not yet been trained on the current dataset $\mathcal{D}_{tr}^r$ (it only performs inference in bilevel training), its predictions may be suboptimal. The KL divergence term encourages the proxy model to mimic the behavior of this "imperfect" LLM, leading to a slight performance degradation. However, this ensures that the proxy model's data preference aligns with that of the LLM, improving the relevance of the selected training data and ultimately boosting the LLM's downstream task performance. An ablation study on the effect of KL divergence loss is presented in Section 6.2.

From round 2 to round 5, the score model is continuously optimized, leading to more accurate sample weight assignments. This, in turn, enhances the proxy model's performance on the weighted training samples, further improving the quality of data selection.

## 5.4 Evaluation Results on the Downstream Tasks

The LLM is continuously trained for 10,000 steps on the selected data in each round. After completing five rounds of training, we evaluate the zero-shot performance of Pythia-410M/1B on various downstream tasks and report the average accuracy along with the standard error for each dataset(see Table 1. Our algorithm consistently outperforms MATES and random selection methods across multiple tasks. For example on 410M setting, BLISS, compared with MATES, improves 2.4% on

Table 1: Comparison of methods on zero-shot evaluation over multiple downstream datasets (410M/1B model, 25B tokens data). Best results are marked bold. The accuracy with standard error is reported based on the lm-evaluation-harness (Gao et al., 2021) implementation.

| Methods (#FLOPs $\times 10^{19}$) | SciQ | ARC-E | ARC-C | LogiQA | OBQA | BoolQ | HellaSwag | PIQA | WinoGrande | Average |
|---|---|---|---|---|---|---|---|---|---|---|
| **410M Setting:** 410M model, 25B tokens | | | | | | | | | | |
| Random (6.35) | 64.1 (1.5) | 40.2 (1.0) | **25.6** (1.3) | 24.7 (1.7) | 29.4 (2.0) | 58.9 (0.9) | 39.7 (0.5) | 67.1 (1.1) | 50.6 (1.4) | 44.5 (1.3) |
| MATES (8.11) | 65.7 (1.5) | 41.5 (1.0) | 25.0 (1.3) | 26.1 (1.7) | **30.8** (2.1) | **60.6** (0.9) | 41.0 (0.5) | 67.8 (1.1) | 51.8 (1.4) | 45.7 (1.4) |
| BLISS (8.08) | **68.1** (1.5) | **42.2** (1.0) | 25.1 (1.3) | **27.3** (1.7) | 29.6 (2.0) | 59.3 (0.9) | **41.2** (0.5) | **68.2** (1.1) | **52.0** (1.4) | **45.9** (1.4) |
| **1B Setting:** 1B model, 25B tokens | | | | | | | | | | |
| Random (17.67) | 65.8 (1.5) | 43.7 (1.0) | 25.6 (1.3) | 27.5 (1.8) | 31.8 (2.1) | 60.2 (0.9) | 43.8 (0.5) | 68.9 (1.1) | 50.7 (1.4) | 46.4 (1.4) |
| MATES (19.97) | 67.3 (1.5) | 44.9 (1.0) | **25.9** (1.3) | **28.7** (1.8) | 32.2 (2.1) | **60.9** (0.9) | 45.3 (0.5) | 69.5 (1.1) | 52.4 (1.4) | 47.5 (1.4) |
| BLISS (8.08) | **69.4** (1.5) | **45.7** (1.0) | 24.8 (1.3) | 25.8 (1.7) | **33.2** (2.1) | 59.8 (0.9) | **47.8** (0.5) | **71.6** (1.1) | **52.9** (1.4) | **47.9** (1.3) |

Table 2: Average evaluation accuracy (15B tokens data) by pretraining 2.8B model with data selected from the 1B model experiment.

| Methods | Round 1 (Random) | Round 2 | Round 3 |
|---|---|---|---|
| MATES | 45.9 (1.3) | 47.4 (1.3) | 47.6 (1.3) |
| BLISS | 45.2 (1.3) | **47.6** (1.3) | **49.0** (1.3) |

Table 3: Average evaluation accuracy of 3 rounds by pretraining Llama-0.5B model. Llama-134M is deployed as the proxy model in BLISS.

| Methods | Round 1 (Random) | Round 2 | Round 3 |
|---|---|---|---|
| MATES | 43.12 (1.27) | 44.53 (1.27) | 45.01 (1.27) |
| BLISS | 43.12 (1.27) | **44.57** (1.27) | **45.65** (1.27) |

SciQ, 0.7% on ARC-E, 0.8% on LogiQA, 0.2% on HellaSwag, 0.4% on PIQA, 0.2% on WinoGrande, and 0.2% on average accuracy (see Table 8). Additionally, Figure 2 presents the evaluation results in relation to pretraining FLOPs and training steps. BLISS consistently outperforms other baseline methods throughout the entire five-round pretraining process (with 10k steps per round). In particular, our method on 1B setting achieves a $1.7\times$ speedup in reaching the same performance as MATES, further validating the effectiveness of our data selection approach.

**Scaling Up to 2.8B Model Pretraining using the Data Selected by 160M/1B Experiment.** To further validate the selected data is of good quality regardless of model size, we pretrain a larger model of 2.8B parameters with data selected from the 1B model experiment with 160M proxy and score models. We run MATES and BLISS for 3 rounds (15B tokens). As shown in Table 2, BLISS consistently outperforms MATES across all data selection rounds, achieving 1.4% accuracy improvement over MATES in round 3.

**Generalize model architecture to LLaMA family.** We also explore LLaMA architecture models to validate the generalization of our method. Specifically, we use LLaMA-0.5B as the target pretraining model, and LLaMA-134M as the proxy model and score model. In each round, we first minimize the difference between the proxy model and the target model by training the proxy model toward a lower KL divergence. Then we periodically reset the proxy model to the initial state, in addition to resetting it at the beginning of each round. Table 3 presents the evaluation results compared with MATES, where BLISS exhibits strong data selection performance. At the round 3, our algorithm improves over MATES by 0.6%. More details are presented in Appendix B.

## 5.5 COMPUTATIONAL COST

We follows the FLOPs estimation method in Li et al. (2024) and report the total GPU FLOPs, including the pretraining, model warm-up, and data selection. Our main observation is: **without relying on any external pretrained models as required in MATES, BLISS achieves higher average downstream performance while consuming fewer FLOPs**. A detailed comparison of total FLOPs consumption is provided in Table 4, and running time/memory comparison is presented in Appendix H.

With the same pretraining budget for LLM and an equivalent number of training tokens, BLISS is more efficient in data selection than MATES. The higher computational cost of MATES is due to its reliance on oracle data influence estimation, which involves computing the loss change after performing a one-step gradient descent update on an individual training sample. This process is highly time-consuming, because it requires per-sample gradient and cannot increase the batch size per GPU. In contrast, BLISS formulates data selection as a bilevel optimization problem, enabling the lightweight score model and proxy model to be trained to convergence within relatively few steps, i.e., 3,000 per round (ablation study for bilevel steps is presented in Appendix I). While BLISS introduces additional training steps for warming up the proxy and score models from scratch, this cost is negligible compared to the overall pretraining FLOPs.

Table 4: Total FLOPs for pretraining 410M/1B model with 25B tokens.

| Model | #FLOPs $\times 10^{19}$ | Ratio | Model | #FLOPs $\times 10^{19}$ | Ratio |
|---|---|---|---|---|---|
| **MATES:** 410M model, 25B tokens | | | **BLISS:** 410M model, 25B tokens | | |
| Model pretraining | 6.35 | 78.3% | Model pretraining | 6.35 | 79.28% |
| Oracle data influence collection | 0.29 | 3.58% | Warm up the proxy/score model | 0.07 | 0.87% |
| Data influence model training | 0.01 | 0.1% | Bilevel optimization | 0.13 | 1.62% |
| Data influence model inference | 1.46 | 18.0% | Data influence model inference | 1.53 | 19.10% |
| **Total** | 8.11 | 100.0% | **Total** | **8.08** | 100.00% |
| **MATES:** 1B model, 25B tokens | | | **BLISS:** 1B model, 25B tokens | | |
| Model pretraining | 17.67 | 88.5% | Model pretraining | 17.67 | 90.48% |
| Oracle data influence collection | 0.83 | 4.1% | Warm up the proxy/score model | 0.07 | 0.36% |
| Data influence model training | 0.01 | 0.1% | Bilevel optimization | 0.261 | 1.34% |
| Data influence model inference | 1.46 | 7.3% | Data influence model inference | 1.53 | 7.83% |
| **Total** | 19.97 | 100.00% | **Total** | **19.53** | 100.00% |

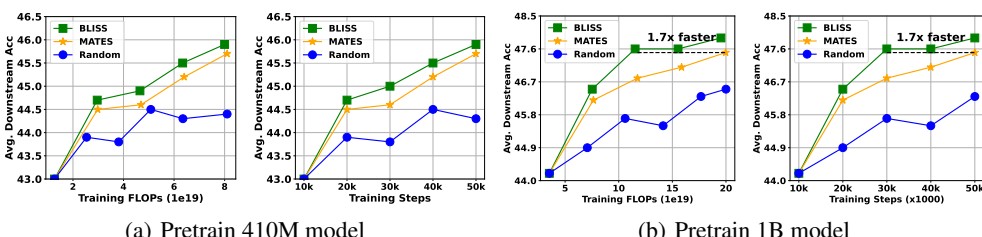

(a) Pretrain 410M model          (b) Pretrain 1B model

Figure 2: The downstream performance of Pythia-410M/1B model w.r.t. pretraining FLOPs and steps, where the first point denotes the performance of a warm-up model trained on random data.

## 6 ABLATION STUDIES

To inspect the effectiveness of key techniques used in our proposed algorithm, we conduct ablation studies on the effect of bilevel optimization (Section 6.1), KL divergence loss (Section 6.2), the impact of softmax reparameterization on the score model's outputs (Appendix C.1), the size of proxy model (Appendix C.2), the initialization for the score model (Appendix C.3), and the influence of different validation datasets ($\mathcal{D}_{val}$) on performance (Appendix C.4).

### 6.1 SINGLE-LEVEL VERSUS BILEVEL OPTIMIZATION

In bilevel algorithm, the hyper-gradient is essential for the update of upper level parameters. To verify the effectiveness of bilevel update for the upper-level parameters, we compare bilevel update with a single update, which update $\theta_s$ and $\theta_p$ together using both training and validation data for the lower-level objective. Specifically, the upper and lower levels are reduced to a single level problem: the upper-level and lower-level parameters are updated simultaneously on validation dataset and training dataset respectively. With the same number of training steps as bilevel training, the average accuracy of single level update degrades $0.5\%$ as shown in Table 5 in Appendix Appendix C.

### 6.2 KL DIVERGENCE ALIGNS THE PROXY MODEL WITH THE LLM

Our objective is to select training data that maximizes the LLM's performance on downstream tasks. To achieve this, the proxy model must effectively represent the LLM, which we enforce by applying KL divergence loss to align their output logits. As shown in Figure 3 (Appendix C), incorporating KL divergence leads to improved performance across most downstream tasks, with a 9.3% accuracy boost on LogiQA and a 1.4% increase in average accuracy. Interestingly, while removing KL divergence results in a lower validation loss (as seen in Figure 7 compared to Figure 6(a) in Appendix F), it does not translate to better downstream performance. These findings highlight the importance of bridging the gap between the proxy model and the LLM to ensure effective data selection, demonstrating that a closer alignment between the two models leads to better overall performance.

## 7 CONCLUSION

In this paper, we present BLISS, a lightweight bilevel influence scoring method for data selection in language model pretraining. BLISS utilizes a proxy model, a score model, and a novel bilevel optimization framework to capture the long-term influence of data without relying on external pretrained models. Experimental results demonstrate its effectiveness in selecting data for pretraining Pythia and LLaMA models. However, current data selection methods primarily focus on language models. In future work, we plan to extend our approach to visual or multimodal models.

## REPRODUCIBILITY STATEMENT

We submit an anonymized code with training/evaluation scripts, configurations, seeds, and environment files in the supplementary materials. All base models are publicly available: Pythia (under EleutherAI Apache-2.0 license) and LLaMA (under Meta Llama 2 Community License Agreement). Datasets C4 is accessible on HuggingFace under the licenses stated on their corresponding Hugging Face dataset cards (loganengstrom/dsdm-candidate-c4). We include download scripts, preprocessing/splits, and references to their dataset cards. These materials sufficiently support the reproduction of our results.

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

## A  DETAILS OF MODEL SETTINGS

The proxy model and score model serve different purposes: the proxy model acts as a surrogate for the LLM and is trained for next-token prediction, while the score model functions as a regression model that maps individual samples to their corresponding influence scores. To transform the proxy model into the score model, we modify its architecture by replacing the final `Linear` layer with an `AdaptiveAvgPool` layer, followed by a `Linear` layer and a `Sigmoid` activation. Specifically, given the output from the preceding transformer blocks with dimension [`Batch,token_size,Emb_size`], the `AdaptiveAvgPool` layer computes the average embedding feature across tokens. The `Linear` layer then maps the pooled token representations to a single-dimensional output, which is subsequently passed through a `Sigmoid` activation to produce an influence score within the range $(0, 1)$. In contrast, the proxy model's final `Linear` layer maps features from previous layers to the vocabulary dimension for token prediction.

## B  IMPLEMENTATION DETAILS IN LLaMA EXPERIMENT

**Model Setup**    In LLaMA setting, the target model is LLaMA-0.5B, and the proxy/score model is LLaMA-134M. They are warmed up under the same process as Pythia setting.

**Training Details of Proxy/Score Model**    There is a little difference in how we deal with the proxy model in LLaMA setting compared to Pythia setting. In addition to resetting the proxy model (LLaMA-134M) at the beginning of each round, we reset it to the initial state every 50 steps of the update of the score model. We distill the target model into the proxy model by minimizing the KL divergence for 240 steps. Then the checkpoint of the proxy model is saved as "initial" state. Since periodic resetting the proxy model ensures a close alignment between two models, we remove the KL divergence regulation term in the lower level loss function. To achieve a better lower-level solution, the proxy model executes 4 lower-level updates, each computed on a batch of 64 samples. After the score model is trained for 50 optimization steps, we reset the proxy model to the initial state.

## C  MORE ABLATION STUDIES

In this section, we provide more ablation studies to verify the effectiveness of each component in our algorithm design.

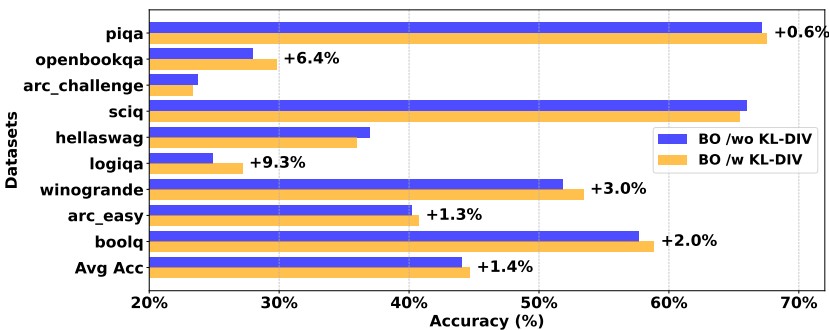

Figure 3: The performance comparison of bilevel optimization with/without KL divergence. The number on the bar indicate the accuracy improvement compared to the method without KL divergence.

Table 5: Comparison of BLISS with different settings(without softmax and single level updata) over multiuple downstream datasets (410M model, 10B tokens) with 20k-step training.

| Methods | SciQ | ARC-E | ARC-C | LogiQA | OBQA | BoolQ | HellaSwag | PIQA | WinoGrande | Average |
|---|---|---|---|---|---|---|---|---|---|---|
| Without softmax | 63.5$_{(1.5)}$ | 41.0$_{(1.0)}$ | 22.4$_{(1.2)}$ | 25.7$_{(1.7)}$ | 30.0$_{(2.1)}$ | 52.8$_{(0.9)}$ | 38.8$_{(0.5)}$ | 67.4$_{(1.1)}$ | 51.0$_{(1.4)}$ | 43.6$_{(1.3)}$ |
| Single Level | 64.4$_{(1.5)}$ | 42.3$_{(1.0)}$ | 22.2$_{(1.2)}$ | 24.1$_{(1.7)}$ | 30.6$_{(2.1)}$ | 55.0$_{(0.9)}$ | 39.7$_{(0.5)}$ | 67.1$_{(1.1)}$ | 52.1$_{(1.4)}$ | 44.2$_{(1.3)}$ |
| BLISS | 65.5$_{(1.5)}$ | 40.8$_{(1.0)}$ | 23.4$_{(1.2)}$ | 27.2$_{(1.7)}$ | 29.8$_{(2.0)}$ | 58.9$_{(0.9)}$ | 36.0$_{(0.5)}$ | 67.6$_{(1.1)}$ | 53.4$_{(1.4)}$ | 44.7$_{(1.3)}$ |

Table 6: Comparison of BLISS with different size of proxy/score model and on zero-shot evaluation over multiuple downstream datasets (410M model, 10B tokens) with 20k-step training.

| Method | SciQ | ARC-E | ARC-C | LogiQA | OBQA | BoolQ | HellaSwag | PIQA | WinoGrande | Average |
|---|---|---|---|---|---|---|---|---|---|---|
| BLISS (Pythia-31M) | 65.5(1.5) | 40.8(1.0) | 23.4(1.2) | 27.2(1.7) | 29.8(2.0) | 58.9(0.9) | 36.0(0.5) | 67.6(1.1) | 53.4(1.4) | 44.7(1.3) |
| BLISS (Pythia-160M) | 63.8(1.5) | 40.8(1.0) | 23.4(1.2) | 27.5(1.8) | 29.8(2.0) | 51.3(0.9) | 38.3(0.5) | 67.6(1.1) | 50.4(1.4) | 44.1(1.3) |

## C.1 SOFTMAX REPARAMETRIZATION FOR SCORE MODEL'S OUTPUT

In our experiment, we apply a softmax function on all batch samples' score across GPUs to obtain the importance weights $P_i$. Note that the raw output of the score model is already within the range $(0, 1)$, but we add another softmax function on top of it. We want to demonstrate the effectiveness of this softmax reparameterization. Intuitively, the main benefit is that it naturally amplifies important samples while downweighting less useful ones, improving the overall data selection process.

To assess the impact of the softmax reparameterization, we conduct an ablation experiment comparing two approaches: (i) naive weighting, where the raw outputs of the score model are used directly as sample weights; (ii) softmax weighting, where the softmax-transformed outputs of the score model determine the sample weights. The results, shown in Table 5, indicate that softmax weighting consistently outperforms naive weighting, leading to a 1.1% improvement in average downstream accuracy. This demonstrates that softmax effectively enhances data selection by better distinguishing important samples.

## C.2 THE SIZE OF PROXY MODEL

We conduct experiments using two different sizes of proxy/score models (31M and 160M) for a 410M LLM. We observe that the KL divergence between the proxy and the LLM remains low for both sizes-0.15 for the 160M model and 0.10 for the 31M model. The corresponding learning curves are shown in Figure 4, which presents the results from round 2. The performance comparison of two sizes of proxy model is summarized in Table 6. These findings suggest that even a small proxy model (31M) is sufficient to serve as an effective surrogate for the 410M LLM.

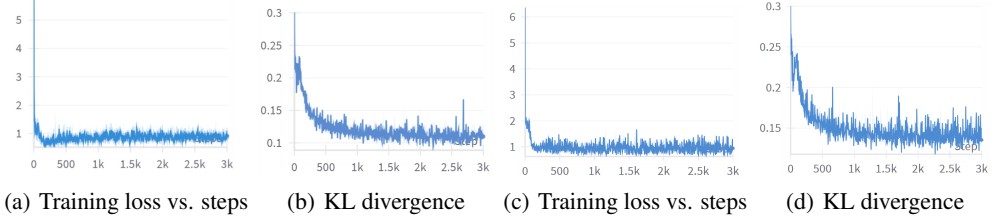

   (a) Training loss vs. steps     (b) KL divergence     (c) Training loss vs. steps     (d) KL divergence

Figure 4: The evolution of the lower-level training loss and KL divergence for different proxy model size. Subfigures (a), (b): Proxy model size 31M, target LLM size 410M. Subfigures (c), (d): Proxy model size 160M, target LLM size 410M.

## C.3 INITIALIZATION METHOD FOR THE SCORE MODEL

In Algorithm 1, we initialize the score model in each new round using the parameters from the last round. This design is motivated by the role of the score model: it learns data representations and ranks the importance of training samples. As training progresses, the model's ability of feature learning improves, making it beneficial to retain learned representations across rounds.

To validate this, we conduct ablation studies comparing two cases:

1. **Original BLISS (BLISS-org)**: the score model in each round is initialized with the parameters from the last round.

2. **Modified Initialization (BLISS$^{\dagger}$)**: the score model in each round is reset to its initial parameters from round 1.

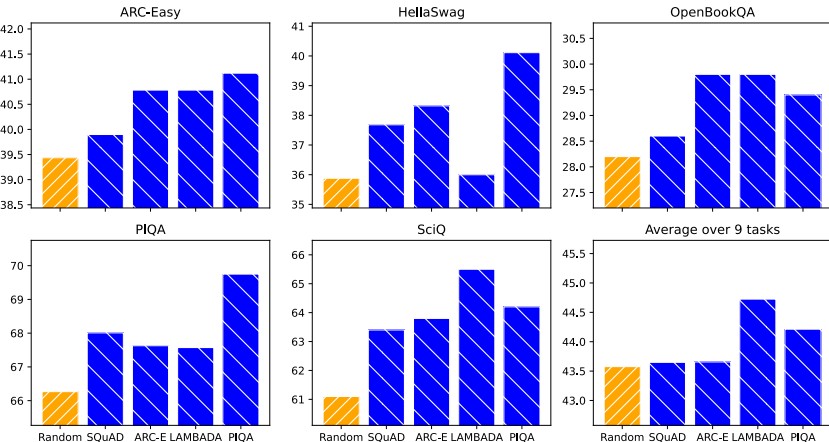

Figure 5: Comparison of BLISS trained with different validation datasets (410M model, 10B tokens). We compare our method with different validation datasets with random selection on 1 downstream task in each subplot.

We then use the trained score models from two cases to select training data and pretrain the target LLM for 15B tokens, respectively. The resulting LLMs are evaluated on multiple downstream datasets. As shown in Table 7, BLISS[†] achieves an average performance that is $0.4\%$ lower than BLISS-org, demonstrating that continuous initialization leads to better data ranking and improved downstream performance.

Table 7: Comparison of methods on zero-shot evaluation over multiple downstream datasets (410M model, 15B tokens). BLISS-org denotes the original algorithm, and BLISS[†] is a variant which uses different initialization method for the score model.

| Methods (#FLOPs $\times 10^{19}$) | SciQ | ARC-E | ARC-C | LogiQA | OBQA | BoolQ | HellaSwag | PIQA | WinoGrande | Average |
|---|---|---|---|---|---|---|---|---|---|---|
| BLISS-org | 67.7 (1.5) | 41.7 (1.0) | 23.6 (1.2) | 25.8 (1.7) | 28.4 (2.0) | 56.0 (0.8) | 39.7 (0.5) | 68.7 (1.1) | 53.2 (1.4) | 44.9 (1.3) |
| BLISS[†] | 65.2 (1.5) | 41.6 (1.0) | 23.4 (1.2) | 27.1 (1.7) | 29.8 (2.0) | 57.5 (0.8) | 34.9 (0.5) | 67.7 (1.1) | 53.5 (1.4) | 44.5 (1.3) |

### C.4 VALIDATION DATASETS

The upper-level optimization aims to minimize the proxy model's loss on the validation dataset, meaning different validation datasets influence data selection. We use different validation set, including SQUAD, ARC-E, LAMBADA, and PIQA, to conduct the bilevel data training, then compare the corresponding downstream performance.

As shown in Figure 5, our algorithm outperforms random selection on most downstream tasks, except BoolQ, regardless of the validation dataset. Notably, LAMBADA yields the highest average accuracy, improving 1.15% over random selection, likely due to its broad domain coverage.

We also notice that our averaged performance is greatly affected by the accuracy of BoolQ task across all validation datasets. This indicates that it is hard to learn when the answer is too short like yes or no.

## D ADDITIONAL RESULTS

Since we use the same experimental settings as MATES(Yu et al., 2024), including pretraining model, data and training steps, we evaluate MATES on the downstream tasks with their checkpoint model (https://huggingface.co/yuzc19/pythia-410m-mates/blob/main/iter-200800-ckpt.pth) of 50k steps. For other baselines, we quote Table 1 from MATES(Yu et al., 2024) for convenience of look-up for the performance of more algorithms.

Table 8: Results of Different Methods under the 410M/1B Setting. Subscripts denote standard deviations. Best scores are in bold.

| Methods$_{(\#FLOPs*1e19)}$ | SciQ | ARC-E | ARC-C | LogiQA | OBQA | BoolQ | HellaSwag | PIQA | WinoGrande | Average |
|---|---|---|---|---|---|---|---|---|---|---|
| **410M Setting:** 410M model, 25B tokens | | | | | | | | | | |
| Random$_{(6.35)}$ | 64.1$_{(1.5)}$ | 40.2$_{(1.0)}$ | **25.6**$_{(1.3)}$ | 24.7$_{(1.7)}$ | 29.4$_{(2.0)}$ | 58.9$_{(0.9)}$ | 39.7$_{(0.5)}$ | 67.1$_{(1.1)}$ | 50.6$_{(1.4)}$ | 44.5$_{(1.3)}$ |
| DSIR$_{(6.35)}$ | 63.1$_{(1.5)}$ | 39.9$_{(1.0)}$ | 23.8$_{(1.2)}$ | 27.0$_{(1.7)}$ | 28.4$_{(2.0)}$ | 58.3$_{(0.9)}$ | 39.6$_{(0.5)}$ | 66.8$_{(1.1)}$ | 51.5$_{(1.4)}$ | 44.3$_{(1.3)}$ |
| LESS$_{(246.35)}$ | 64.6$_{(1.5)}$ | 42.3$_{(1.0)}$ | 23.1$_{(1.2)}$ | 25.2$_{(1.7)}$ | 30.4$_{(2.1)}$ | 55.6$_{(0.9)}$ | **41.9**$_{(0.5)}$ | 67.2$_{(1.1)}$ | 51.0$_{(1.4)}$ | 44.6$_{(1.4)}$ |
| SemDeDup$_{(7.81)}$ | 63.5$_{(1.5)}$ | **42.4**$_{(1.0)}$ | 24.4$_{(1.3)}$ | **27.6**$_{(1.7)}$ | 30.0$_{(2.1)}$ | 58.2$_{(0.9)}$ | 40.8$_{(0.5)}$ | 67.8$_{(1.1)}$ | 52.3$_{(1.4)}$ | 45.2$_{(1.3)}$ |
| DsDm$_{(10.72)}$ | 65.4$_{(1.5)}$ | 41.7$_{(1.0)}$ | 24.7$_{(1.3)}$ | 27.5$_{(1.8)}$ | 29.0$_{(2.1)}$ | 57.5$_{(0.9)}$ | 40.3$_{(0.5)}$ | 67.1$_{(1.1)}$ | 50.1$_{(1.4)}$ | 44.9$_{(1.4)}$ |
| QuRating$_{(26.35)}$ | 64.8$_{(1.5)}$ | 42.0$_{(1.0)}$ | 25.4$_{(1.3)}$ | 25.3$_{(1.7)}$ | 30.2$_{(2.1)}$ | 58.9$_{(0.9)}$ | 40.7$_{(0.5)}$ | 67.5$_{(1.1)}$ | 52.1$_{(1.4)}$ | 45.2$_{(1.4)}$ |
| MATES$_{(8.11)}$ | 65.7$_{(1.5)}$ | 41.5$_{(1.0)}$ | 25.0$_{(1.3)}$ | 26.1$_{(1.7)}$ | **30.8**$_{(2.1)}$ | **60.6**$_{(0.9)}$ | 41.0$_{(0.5)}$ | 67.8$_{(1.1)}$ | 51.8$_{(1.4)}$ | 45.7$_{(1.4)}$ |
| BLISS$_{(8.08)}$ | **68.1**$_{(1.5)}$ | 42.2$_{(1.0)}$ | 25.1$_{(1.3)}$ | 27.3$_{(1.7)}$ | 29.6$_{(2.0)}$ | 59.3$_{(0.9)}$ | 41.2$_{(0.5)}$ | **68.2**$_{(1.1)}$ | **52.0**$_{(1.4)}$ | **45.9**$_{(1.4)}$ |
| **1B Setting:** 1B model, 25B tokens | | | | | | | | | | |
| Random$_{(17.67)}$ | 65.8$_{(1.5)}$ | 43.7$_{(1.0)}$ | 25.6$_{(1.3)}$ | 27.5$_{(1.8)}$ | 31.8$_{(2.1)}$ | 60.2$_{(0.9)}$ | 43.8$_{(0.5)}$ | 68.9$_{(1.1)}$ | 50.7$_{(1.4)}$ | 46.4$_{(1.4)}$ |
| DSIR$_{(17.67)}$ | 65.8$_{(1.5)}$ | 42.6$_{(1.0)}$ | 24.7$_{(1.3)}$ | **28.7**$_{(1.8)}$ | 29.2$_{(2.0)}$ | 59.7$_{(0.9)}$ | 44.2$_{(0.5)}$ | 68.3$_{(1.1)}$ | **53.2**$_{(1.4)}$ | 46.3$_{(1.4)}$ |
| SemDeDup$_{(19.13)}$ | 66.8$_{(1.5)}$ | 45.5$_{(1.0)}$ | 25.3$_{(1.3)}$ | 27.6$_{(1.8)}$ | 30.6$_{(2.1)}$ | 60.2$_{(0.9)}$ | 45.3$_{(0.5)}$ | 69.7$_{(1.1)}$ | 52.5$_{(1.4)}$ | 47.1$_{(1.4)}$ |
| DsDm$_{(22.04)}$ | 68.2$_{(1.5)}$ | 45.0$_{(1.0)}$ | **26.5**$_{(1.3)}$ | 26.6$_{(1.7)}$ | 29.4$_{(2.0)}$ | 59.0$_{(0.9)}$ | 44.8$_{(0.5)}$ | 68.9$_{(1.1)}$ | 51.9$_{(1.4)}$ | 46.7$_{(1.3)}$ |
| QuRating$_{(37.67)}$ | 67.1$_{(1.5)}$ | 45.5$_{(1.0)}$ | 25.6$_{(1.3)}$ | 26.9$_{(1.7)}$ | 29.8$_{(2.0)}$ | 60.3$_{(0.9)}$ | 45.2$_{(0.5)}$ | 70.2$_{(1.1)}$ | 51.6$_{(1.4)}$ | 46.9$_{(1.3)}$ |
| MATES$_{(19.97)}$ | 67.3$_{(1.5)}$ | 44.9$_{(1.0)}$ | 25.9$_{(1.3)}$ | **28.7**$_{(1.8)}$ | 32.2$_{(2.1)}$ | **60.9**$_{(0.9)}$ | 45.3$_{(0.5)}$ | 69.5$_{(1.1)}$ | 52.4$_{(1.4)}$ | 47.5$_{(1.4)}$ |
| BLISS$_{(8.08)}$ | **69.4**$_{(1.5)}$ | **45.7**$_{(1.0)}$ | 24.8$_{(1.3)}$ | 25.8$_{(1.7)}$ | **33.2**$_{(2.1)}$ | 59.8$_{(0.9)}$ | **47.8**$_{(0.5)}$ | **71.6**$_{(1.1)}$ | 52.9$_{(1.4)}$ | **47.9**$_{(1.3)}$ |

Table 9: Experimental settings.

| Hyperparameters | Values |
|---|---|
| *Pretrain* | |
| Data set | C4 |
| Tokens | 25B |
| Model | Pythia-410M/1B/2.8B, LLaMA-0.5B |
| batch size | 512 |
| Sequence length | 1024 |
| Max learning rate | 1e-3 |
| *bilevel optimization* | |
| Proxy/Score model | Pythia-31M (for 410M LLM), Pythia-160M (for 1B LLM), LLaMA-134M (for LLaMA-0.5B LLM) |
| $\gamma$ | 1e-2 |
| $\lambda$ | 1e-6 |
| batch size | 16(Pythia-410M, LLaMA-0.5B)/32(Pythia 1B) |
| Proxy/Score model learning rate($\eta_1/\eta_3$) | 1e-5 |
| GDLS learning rate($\eta_2$) | 1e-2 |
| GDLS steps($K$) | 3 |
| Score model steps | 3k(Pythia-410M/1B)/1k(LLaMA-0.5B) |
| Proxy model steps | 3k(Pythia-410M/1B)/1k(LLaMA-0.5B) |
| Initialization of score/proxy model | Randomly initialized |

# E    EXPERIMENTAL HYPERPARAMETERS

Table 9 shows the hyperparameter settings in our experiments. We use cosine learning rate scheduler in bilevel optimization, WSD(Yu et al., 2024) learning rate scheduler for pretraining and constant learning rate for GDLS. We use double loop to update the proxy model when employing 1B LLM, i.e., 5 steps for the lower level update. The experiments run on 8 A6000 GPUs with Distributed Data Parallel (DDP) strategy.

# F    EVOLUTION OF TRAINING AND VALIDATION LOSS

In Figure 6(a), 6(b), we visualize the curves training loss pretraining round 2 and 5.

# G    DISTRIBUTED SOFTMAX TO COMPUTE INFLUENCE SCORE

In bilevel optimization, the importance weight $P_i$ is computed based on a mini batch that is distributed across different GPUs. However, back propagation through different GPUs is not implemented by Pytorch. Thus we deploy "distributed softmax" in practice. In detail, our implementation requires 3

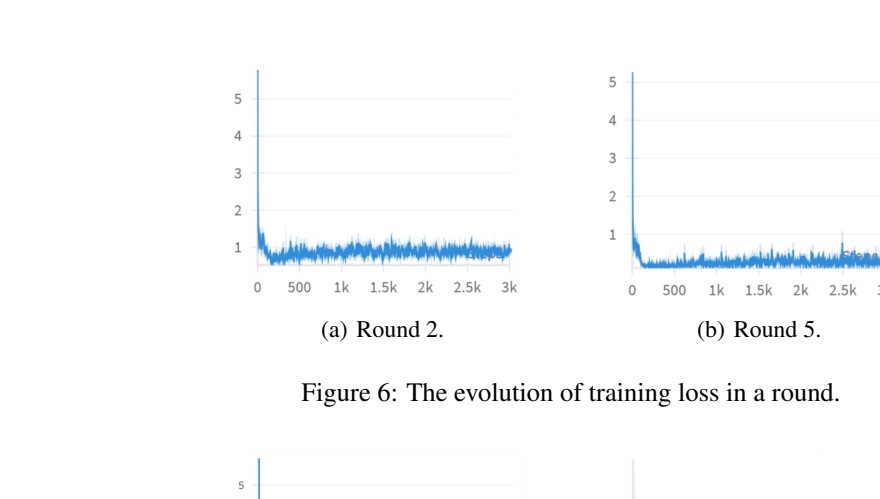

(a) Round 2.                           (b) Round 5.

Figure 6: The evolution of training loss in a round.

(a) Training loss vs. steps            (b) Validation loss vs. steps

Figure 7: The visualization of lower-level training loss and the upper-level validation loss in round 2 bilevel optimization without KL divergence.

times of communication among GPUs.

$$P_i = \frac{e^{h(\theta_s;\xi_i)}}{\sum_{j=1}^{B} e^{h(\theta_s;\xi_j)}} \qquad (6)$$

As equation (6) shows, the denominator of $P_i$ is the summation of every sample's exponential score. Therefore, in the first communication, each GPU gets the scores from others and calculates the denominator locally. A second communication is required to compute the term $\sum_{j=1}^{\mathcal{B}} P_j \nabla_{\theta_s} h(\theta_s^t; \xi_j)$ in equation (5). In detail, we need to gather gradients of $h$ and $\mathcal{L}$' of every sample across all GPUs. After computing hyper-gradients of every sample, they are accumulated to update upper-level variables. With efficient communication API provided by Fabric `https://lightning.ai/docs/fabric/stable/`, the time consumed in bilevel optimization of each round is within 1.5 hours.

## H   RUNNING TIME AND MEMORY

We measured the memory and runtime of the data selection stage for both BLISS and MATES under different target (or pretraining) model sizes (for short, T: target). The results are shown in Table 10. We have two observations:

- BLISS scales well with larger target models. Note that the target model is not an optimization variable for the bilevel optimization and it is only used for calculating the KL divergence. Therefore, it does not affect the scalability of bilevel optimization. When increasing the target from 410M to 1B, BLISS's memory and runtime grow moderately ($49.46 \rightarrow 74.51$ GB; $5.03 \rightarrow 11.82$ hours), as expected.

- BLISS is significantly faster than MATES. MATES incurs high cost because each round requires oracle data collection. For every example, MATES performs a one-step gradient update on the target model and evaluates the validation loss change to compute influence scores. This per-example simulation dominates runtime. In contrast, BLISS avoids all

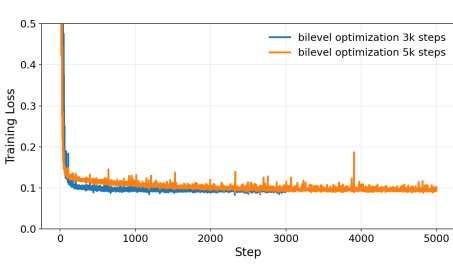

Figure 8: Training loss with different steps.

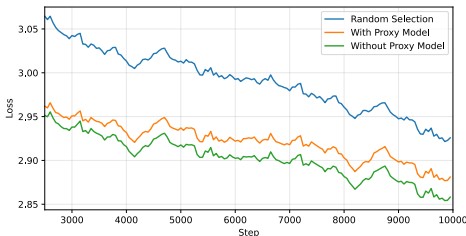

Figure 9: Test loss on SlimPajama-6B.

per-example oracle evaluations in MATES and performs bilevel optimization directly on the proxy/score model, leading to $2 - 3\times$ faster data selection.

Table 10: The comparison of runtime and memory in data selection stage.

| Setting | Memory peak (GB) | Total data selection time (hours) |
|---|---|---|
| MATES: T: 410M | 36.36 | 18.0 |
| MATES: T: 1B | 63.52 | 30.32 |
| BLISS: T: 410M | 49.46 | 5.03 |
| BLISS: T: 1B | 74.51 | 11.82 |

## I    ABLATION STUDY FOR BILEVEL OPTIMIZATION STEPS

We did the ablation study to investigate the steps of bilevel optimization. As shown in Figure 8, both the 3k-step and 5k-step settings converge to nearly the same training loss. This indicates that 3k steps are sufficient for the proxy model, as increasing the steps to 5k does not yield additional improvements. So we fix the training steps of bilevel optimization to 3k steps in main experiments.

## J    DOMAIN REWEIGHTING

To verify the fidelity of proxy models to full-scale LLMs, we conduct a domain-reweighting experiment on the SlimPajama-6B dataset (DKYoon, 2023), which contains $d = 7$ domains: ArXiv, Books, C4, CommonCrawl, GitHub, StackExchange, and Wikipedia. The objective is to learn optimal domain weights $\alpha \in \mathbb{R}^d$ such that a model trained on data sampled according to the weights achieves the best downstream performance.

We compare two settings:

1. **Case 1 (with proxy model):** The lower level optimizes a lightweight proxy model (LLaMA-134M) with output alignment to the target LLM (LLaMA-300M), and the upper level learns the domain weights $\tilde{\alpha}$.

2. **Case 2 (without proxy model):** The lower level directly optimizes the target LLM (LLaMA-300M), and the upper level learns the domain weights $\alpha$.

We perform bilevel optimization for 1,000 steps in both cases to learn the domain weights, where 10% of the original training set is held out as the validation set for the upper-level objective, and the remaining 90% is used as the lower-level training set. After obtaining $\tilde{\alpha}$ and $\alpha$, we train two final LLaMA-300M models on data sampled according to each set of weights, respectively . Figure 10 presents the learning curves of domain weights for both cases. We observe that the trajectories of $\tilde{\alpha}$ and $\alpha$ are highly similar across most domains (e.g., in the domain of Wikipedia, Book, Stackexchange), demonstrating that the proxy model maintains high fidelity to the full-scale LLM in data selection.

Finally, we evaluate the resulting pretrained LLMs on the test set of SlimPajama-6B, and the results are shown in Figure 9. The test loss curves show that data selection based on the proxy model maintains high fidelity to the full-scale LLM, while significantly outperforming random selection.

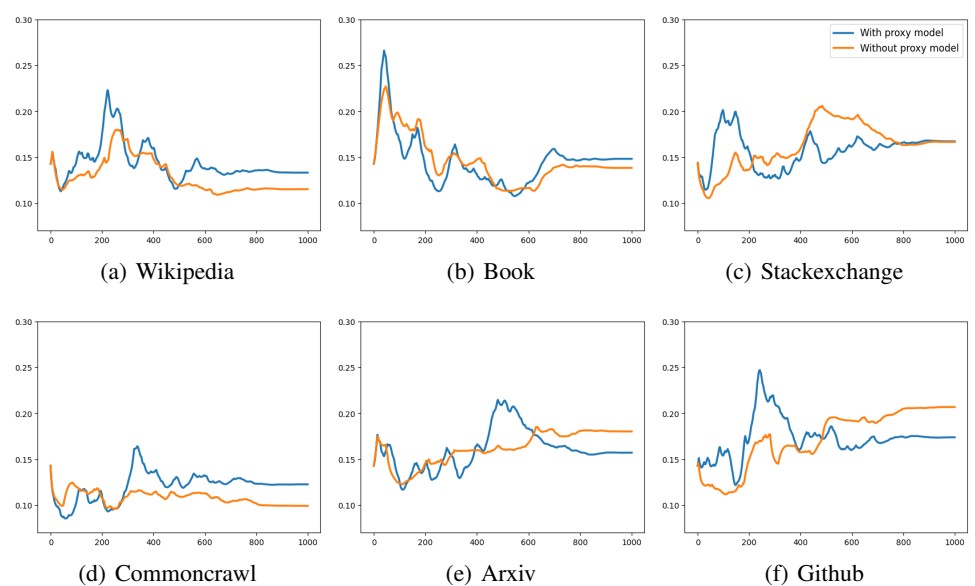

Figure 10: Learning curves of domain weights.

## K    THE USE OF LARGE LANGUAGE MODELS (LLMs)

LLMs are not involved in our research methodology. Their use is limited to polish the writing.