# OpenReview forum: "BLISS: A Lightweight Bilevel Influence Scoring Method for Data Selection in Language Model Pretraining"
_ICLR.cc/2026/Conference — Submitted to ICLR 2026_

### Official Review · Reviewer_X4zu · 2025-10-19

**Soundness:** 2
**Presentation:** 2
**Contribution:** 2
**Rating:** 4
**Confidence:** 4

**Summary:**

This paper proposes BLISS, a data selection method for llm pre-training. Specifically, authors formulate data selection as a bilevel optimization problem, targeting at maximizing llm performance on validation sets. Experiments on C4 dataset and a series of 410M/1B/2.8B llms validate the effectiveness of BLISS.

**Strengths:**

- The research topic of this paper is timely and important.
- This paper is relatively well-written.

**Weaknesses:**

1. The paper heavily emphasizes operating "from scratch" without external pretrained models, yet still requires a warm-up phase using randomly selected data (Section 4.3). Additionally, the bilevel optimization lower-level objective includes KL divergence with a specific LLM, which must already be partially trained to provide meaningful output logits. This fundamentally contradicts the core claim of independence from pretrained models.

2. The paper claims to capture "long-term impact" by training the proxy model "to convergence" (Equation 1). However, the proxy model is trained for only 3,000 steps per round and is reset to initial parameters at the beginning of each round (line 4, Algorithm 1). This is not convergence in any meaningful sense, and resetting eliminates any accumulated long-term knowledge. Authors should provide empirical evidence (e.g., loss curves, model performance) demonstrating that 3,000 steps represents convergence, or revise claims about capturing "long-term" vs. "medium-term" impact.

3. Although the paper presents a bilevel optimization formulation but provides no convergence guarantees, no sample complexity analysis, and no theoretical justification for why this formulation should select better data than alternatives. Without theory, it's unclear whether observed improvements are due to the principled framework or implementation details. At least, authors should add theoretical analysis showing: (a) convergence rates of the bilevel optimization algorithm, (b) bounds on the approximation error of using proxy models, (c) conditions under which the selected data provably improves downstream performance.

4. Table 4 shows BLISS uses 8.08×10^19 FLOPs vs. MATES' 8.11×10^19 for 410M models, claiming efficiency. However, this comparison is not reasonable because

- MATES trains per-sample gradients (inherently expensive) while BLISS batches operations differently.
- The comparison doesn't account for memory overhead of maintaining two additional models.
- Wall-clock time comparisons are absent.
- Different parallelization strategies could dramatically affect practical efficiency.

5. Many reported improvements fall within standard error margins. For example, in Table 1 (410M setting):

- SciQ: 68.1±1.5 (BLISS) vs. 65.7±1.5 (MATES), this is potentially overlapping
- ARC-C: 25.1±1.3 (BLISS) vs. 25.0±1.3 (MATES), this is essentially identical
- Average: 45.9±1.4 vs. 45.7±1.4, this is a marginal difference

6. The paper uses Pythia models throughout (proxy, score, and target LLM) but only briefly tests LLaMA (Appendix B) with different training procedures (periodic resets, KL divergence removal). No analysis examines what happens when proxy and target architectures differ (e.g., Pythia proxy for LLaMA target, or vice versa).

7.  More recent data selection methods aren't compared.
- These are highly relevant methods but are not compared:   DataComp-LM[1],  QuRating[2], Meta-rater[3], QUAD[4].

- Also, it is understandable that works like  [2-3] use external LLMs which costs different FLOPs than training from scratch, making direct cost comparison problematic. However, this paper criticizes methods using external models but doesn't fairly account for when this might be practically cheaper than bilevel optimization.


[1] https://arxiv.org/abs/2406.11794

[2] https://arxiv.org/abs/2402.09739

[3] https://arxiv.org/abs/2504.14194

[4] https://arxiv.org/abs/2409.16986

**Questions:**

Please see weaknesses.

---

> ### Author Response · Authors · 2025-11-27
>
> **Q1. The paper heavily emphasizes operating "from scratch" without external pretrained models, yet still requires a warm-up phase using randomly selected data (Section 4.3). Additionally, the bilevel optimization lower-level objective includes KL divergence with a specific LLM, which must already be partially trained to provide meaningful output logits. This fundamentally contradicts the core claim of independence from pretrained models.**
>
> **A1.** Our claim of “from scratch” refers specifically to not relying on any externally pretrained commercial or open-source LLMs. All models used in BLISS, including the warm-up model and the target model used in the KL term, are pretrained entirely by us within the same compute budget and training pipeline, and are not external checkpoints. The reference model in KL  term is the same model we are training, not a third-party pretrained LLM.
>
> **Q2. The paper claims to capture "long-term impact" by training the proxy model "to convergence" (Equation 1). However, the proxy model is trained for only 3,000 steps per round and is reset to initial parameters at the beginning of each round (line 4, Algorithm 1). This is not convergence in any meaningful sense, and resetting eliminates any accumulated long-term knowledge. Authors should provide empirical evidence (e.g., loss curves, model performance) demonstrating that 3,000 steps represents convergence, or revise claims about capturing "long-term" vs. "medium-term" impact.**
>
> **A2.** We thank the reviewer for pointing out the ambiguity around the term “long-term impact.” In our paper, “long-term” does not refer to training the proxy model to full convergence; rather, it refers to capturing the cumulative influence of data across multiple optimization steps in the current round, instead of the single-step influence commonly used in short-term influence methods. Our proxy model is trained for 3,000 steps per round because this already produces a stable training loss as shown in Figure 6,7 in Appendix F. Resetting the proxy model at each round is intentional: it ensures that the proxy focuses solely on the current round’s data selection. Our pretraining data are partitioned by rounds, and the target LLM is also trained round-by-round. Resetting ensures that the proxy model captures the impact of the current round’s data on the current stage of LLM training, rather than mixing influences across rounds.
>
> **Q3. Although the paper presents a bilevel optimization formulation but provides no convergence guarantees, no sample complexity analysis, and no theoretical justification for why this formulation should select better data than alternatives. Without theory, it's unclear whether observed improvements are due to the principled framework or implementation details. At least, authors should add theoretical analysis showing: (a) convergence rates of the bilevel optimization algorithm, (b) bounds on the approximation error of using proxy models, (c) conditions under which the selected data provably improves downstream performance.**
>
> **A3.** We thank the reviewer for pointing this out. Our algorithm fits into the standard bilevel optimization setting studied in prior work such as [1]. Under the standard assumptions as their paper, one can adapt the analysis of SLIP (Theorem 4.1 in [1] to our algorithm and it converges to $\epsilon$-stationary points, i.e., $\frac{1}{T}\sum_{t=1}^{K}\mathbb{E}\\|\nabla\Phi(\theta_s^{t})\\|\leq \epsilon$ within $\widetilde{O}(\epsilon^{-4})$ (up to logarithmic factors of $1/\epsilon$) iterations in expectation. This result matches the known rates for stochastic bilevel methods.
>
>
> [1] Xiaochuan Gong, Jie Hao, and Mingrui Liu. A nearly optimal single loop algorithm for stochastic
> bilevel optimization under unbounded smoothness. In Forty-first International Conference on
> Machine Learning, 2024.

---

> ### Author Response · Authors · 2025-11-27
>
> **Q4. Table 4 shows BLISS uses $8.08\times10^{19}$ FLOPs vs. MATES' $8.11\times10^{19}$ for 410M models, claiming efficiency. However, this comparison is not reasonable.**
>
> **A4.** We measured the memory and runtime of the data selection stage for both BLISS and MATES under different target (or pretraining) model sizes (for short, T: target). The results are shown in Table 1. We have two observations:
>
> - BLISS scales well with larger target models. Note that the target model is not an optimization variable for the bilevel optimization and it is only  used  for calculating the KL divergence. Therefore, it does not affect the scalability of bilevel optimization. When increasing the target from 410M to 1B, BLISS’s memory and runtime grow moderately ($49.46\rightarrow 74.51$ GB; $5.03\rightarrow 11.82$ hours), as expected.
> - BLISS is significantly faster than MATES. MATES incurs high cost because each round requires oracle data collection. For every example, MATES performs a one-step gradient update on the target model and evaluates the validation loss change to compute influence scores. This per-example simulation dominates runtime. In contrast, BLISS avoids all per-example oracle evaluations in MATES and performs bilevel optimization directly on the proxy/score model, leading to $2-3 \times$ faster data selection.
>
> These results confirm that BLISS introduces modest overhead and scales well to billion-parameter settings. We have included the comparison of running time and memory in Appendix I.
>
> **Table1: Comparison of runtime and memory in the data selection stage.**
> | Setting             | Memory peak (GB) | Total data selection time (hours) |
> |---------------------|------------------|------------------------------------|
> | MATES: T: 410M      | 36.36            | 18.0                               |
> | MATES: T: 1B        | 63.52            | 30.32                              |
> | BLISS: T: 410M      | 49.46            | 5.03                               |
> | BLISS: T: 1B        | 74.51            | 11.82                              |
>
> **Q5. Many reported improvements fall within standard error margins.**
>
> **A5.** Notably, BLISS shows a significant improvement over MATES in the 2.8B-scale pretraining setting (in Table 2 of the original submisson), i.e., BLISS achieves 49.0 vs. 47.6. This suggests that the benefit of BLISS becomes more significant as the pretraining model scales up. Due to computational constraints, 2.8B model is the largest model we can feasibly pretrain from scratch. We believe that with larger compute, scaling BLISS to bigger models would yield larger gains, consistent with the trend observed at 2.8B.
>
> **Q6. The paper uses Pythia models throughout (proxy, score, and target LLM) but only briefly tests LLaMA (Appendix B) with different training procedures (periodic resets, KL divergence removal). No analysis examines what happens when proxy and target architectures differ (e.g., Pythia proxy for LLaMA target, or vice versa).**
>
> **A6.** The purpose of the proxy model is to approximate the behavior of the target model during bilevel optimization. For this reason, we choose a proxy that shares the same architecture family as the target model but at a smaller scale. This structural compatibility ensures that the selected data reflect what the target model prefers, enabling the selected data to improve the target model’s performance as much as possible. If the proxy and target models came from entirely different architectures or families, this behavioral alignment would break, and there would be no guarantee that the selected data would be optimal for the target model. We will clarify this point in the revision. Investigating how to extend BLISS to proxy and target models with different architectures is indeed an important question, and we identify this as a key direction for future research.
>
> We additionally conducted an experiment where the proxy and target models use different architectures. When using Pythia-124M as the proxy model for a LLaMA-0.5B target model, the average downstream accuracy is 43.83 in Round 2. In contrast, when the proxy model is LLaMA-134M, i.e., sharing the same architecture family with the target model, the accuracy improves to 44.57. This result suggests that architecture alignment between the proxy and target LLMs leads to selecting more data that the target model prefers, supporting our design choice of using same-family proxy models.

---

> ### Author Response · Authors · 2025-11-27
>
> **Q7. More recent data [1,2,3,4] selection methods aren't compared.**
>
> **A7.** DataComp-LM [1], QuRating [2], and Meta-rater [3] rely on external pretrained LLMs (either for scoring, filtering, or pairwise comparison), whereas BLISS is explicitly designed for the from-scratch, compute-limited setting where such external models cannot be used. Because their pipelines, compute assumptions, and cost models differ substantially from our problem setup, a direct empirical comparison would not be fair.
> In addition, the code of QUAD [4] is no longer available. Due to time constraints and the complexity of reproducing their pipeline, we were unable to include this comparison.
> We have cited these work and included the discussion of these differences in the related work section to clarify the relationship between BLISS and these methods.
>
> [1] J Li, A Fang, G Smyrnis, M Ivgi, M Jordan, S Gadre, H Bansal, E Guha, S Keh, K Arora, et al. Datacomp-lm: In search of the next generation of training sets for language models, 2024. URL https://arxiv. org/abs/2406.11794.
>
> [2] Alexander Wettig, Aatmik Gupta, Saumya Malik, and Danqi Chen. Qurating: Selecting high-quality data for training language models. arXiv preprint arXiv:2402.09739, 2024.
>
> [3] Xinlin Zhuang, Jiahui Peng, Ren Ma, Yinfan Wang, Tianyi Bai, Xingjian Wei, Qiu Jiantao, Chi Zhang, Ying Qian, and Conghui He. Meta-rater: A multi-dimensional data selection method for pre-training language models. In Proceedings of the 63rd Annual Meeting of the Association for Computational Linguistics (Volume 1: Long Papers), pp. 10856–10896, 2025.
>
> [4] C Zhang, H Zhong, K Zhang, C Chai, R Wang, X Zhuang, T Bai, J Qiu, L Cao, J Fan, et al. Harnessing diversity for important data selection in pretraining large language models. arxiv 2024. arXiv preprint arXiv:2409.16986

---

### Official Review · Reviewer_Uq2j · 2025-10-26

**Soundness:** 3
**Presentation:** 2
**Contribution:** 2
**Rating:** 4
**Confidence:** 3

**Summary:**

This paper introduces BLISS (Bilevel Influence Scoring Method for Data Selection), a lightweight data selection approach for language model pretraining, aiming to address the limitations of existing methods.

Existing data selection methods often rely on external pretrained models, making it hard to isolate the effects of data selection from those of external models, while also neglecting the long-term impact of selected data when the model is trained to convergence. BLISS tackles these issues through a novel bilevel optimization framework: it uses a small proxy model as a surrogate for the large language model (LLM) and a score model to estimate the long-term influence of training samples. The upper-level objective optimizes the score model to assign importance weights to samples, ensuring that the lower-level objective (training the proxy model to convergence on weighted training loss) achieves optimal validation performance. After optimization, the score model predicts influence scores for the dataset, enabling efficient selection of high-quality samples.

**Strengths:**

1. The existing data selection approach (especially Qurating) depend on large external models (e.g., GPT-3.5), which introduces cost, bias, and reproducibility issues. BLISS avoids this entirely.

2. Rather than estimating sample influence based on short-term updates, BLISS uses a proxy model trained to convergence, offering more reliable evaluation of data utility.

3. The use of lightweight proxy and score models significantly reduces computational cost. The total training FLOPs are lower than baseline methods like MATES. Also, across a range of benchmarks (e.g., SciQ, ARC-E, LogiQA), BLISS consistently outperforms baselines, showing generalization benefits from better data selection.

4. Please explain the motivation for performing data selection to retain a small amount of high-quality data during the pretraining phase, rather than preserving a large amount of clean data (e.g. preserving 70%-80% of the full candidate dataset). For details, please refer to the weaknesses section. Also, please show the training performance when using 10%, 20%, ..., up to 70% of the entire candidate data pool selected by BLISS, as well as when using 70% of clean data obtained through a simple filtering method. This would be very interesting.

**Weaknesses:**

1.  I highly commend the authors for providing standard errors but based on Tables 1 and 2, it appears that the improvements on average task accuracy (over MATES) are well within these reported errors? The only result that seems to not be is round 3 for 2.8B scale. Also, for the topic of "pre-training", the experiment scale in this paper is not sufficient. How well the BLISS method generalize to larger model is questionable. When the model becomes large enough, do we still need to select a subset of good examples instead of focusing on getting more high quality data? Also, I think the goal of 'pre-training' is to obtain a strong foundation (unaligned) model that can serve well for later-stage post-training or alignment. However, BLISS introduces a manually specified validation set, which may affect the model's generalization to other tasks. The validation set is more appropriately introduced during the SFT (supervised fine-tuning) or continued pretraining stages, rather than during the pretraining stage.

2. Related to the problem setting, in real-world, when will people do 3B-scale (even 7B-scale) model pre-training from scratch? Usually people either start from a pre-trained checkpoint then do post-training/alignment (for a few specific tasks), or do continuous pre-training/tailpatch training (for knowledge enhancement). If we really need a small scale pre-train ckpt, most likely we will distill a large pre-train ckpt to a smaller model. So how useful this problem setting should be further discussed.

3. Another limitation of this study is that the authors do not fully motivate the problem setting. While cleaning noisy data and removing near-duplicate examples are useful for LLM pretraining, I am not fully convinced why you have to select a subset of “quality data” from the full dataset. For example, if we do a coarse-grained filtering of the original dataset to leave around 70%-80% data and use all of them to do LLM pretraining, will the model performance better.

**Questions:**

1. Can you try combining the validation tasks (e.g. take an average of their performances) for the purposes of targeting?

2. Looking at the MATES paper, it seems you report the same exact numbers for their method. Are you using the exact same training implementation as they did? (Mentioned in NIPS)

3. An assumption for the BLISS approach is that you want to model influence based on training to convergence. However, in practice, most LLMs are pre-trained based upon a finite compute budget rather than true “convergence” (due to the sheer quantities of data involved). Perhaps the authors could comment on this potential mismatch?

---

> ### Author Response · Authors · 2025-11-27
>
> **Q1. I highly commend the authors for providing standard errors but based on Tables 1 and 2, it appears that the improvements on average task accuracy (over MATES) are well within these reported errors? The only result that seems to not be is round 3 for 2.8B scale. Also, for the topic of "pre-training", the experiment scale in this paper is not sufficient. How well the BLISS method generalize to larger model is questionable. When the model becomes large enough, do we still need to select a subset of good examples instead of focusing on getting more high quality data? Also, I think the goal of 'pre-training' is to obtain a strong foundation (unaligned) model that can serve well for later-stage post-training or alignment. However, BLISS introduces a manually specified validation set, which may affect the model's generalization to other tasks. The validation set is more appropriately introduced during the SFT (supervised fine-tuning) or continued pretraining stages, rather than during the pretraining stage.**
>
> **A1.** (1) Experiment scale: Our experimental scale is limited by the hardware: on 4 A100s (80GB memory) with micro-batch 128, training a 2.8B model already saturates memory. Notably, prior data selection works also operate at similar or smaller scales (e.g., DoGE [1] at 684M, MATES [2] at 1B). BLISS itself is a data selection framework, and its proxy-model design keeps all heavy hypergeradient calculation in bilevel optimization off the target LLM. That makes our method be able to scalable to even larger pretraining models.
>
> (2) Motivation: A key motivation of our work is the practical setting where pretraining compute is limited, meaning that training on all available data is often infeasible. In such cases, the central question becomes how to select the most beneficial subset of data to maximize downstream performance under this fixed budget. BLISS is designed specifically for this scenario, which is consistent with prior work [4, 3, 2].
>
>
> (3) Validation set: the downstream task is generally unavailable in the pretraining stage, so the validation set serves only as a signal for selecting data  which can maximize the model performance on the downstream task. This validation set is also widely used in data selection for pretraining [2, 1, 5].
>
> [1] Simin Fan, Matteo Pagliardini, and Martin Jaggi. Doge: Domain reweighting with generalization estimation. arXiv preprint arXiv:2310.15393, 2023.
>
> [2] Zichun Yu, Spandan Das, and Chenyan Xiong. Mates: Model-aware data selection for efficient pretraining with data influence models. arXiv preprint arXiv:2406.06046, 2024.
>
> [3] Logan Engstrom, Axel Feldmann, and Aleksander Madry. Dsdm: Model-aware dataset selection with datamodels. arXiv preprint arXiv:2401.12926, 2024.
>
> [4] Sung Min Park, Kristian Georgiev, Andrew Ilyas, Guillaume Leclerc, and Aleksander Madry. Trak: Attributing model behavior at scale. arXiv preprint arXiv:2303.14186, 2023.
>
> [5] C Zhang, H Zhong, K Zhang, C Chai, R Wang, X Zhuang, T Bai, J Qiu, L Cao, J Fan, et al. Harnessing diversity for important data selection in pretraining large language models. arxiv 2024. arXiv preprint arXiv:2409.16986.

---

> ### Author Response · Authors · 2025-11-27
>
> **Q2. Related to the problem setting, in real-world, when will people do 3B-scale (even 7B-scale) model pre-training from scratch? Usually people either start from a pre-trained checkpoint then do post-training/alignment (for a few specific tasks), or do continuous pre-training/tailpatch training (for knowledge enhancement). If we really need a small scale pre-train ckpt, most likely we will distill a large pre-train ckpt to a smaller model. So how useful this problem setting should be further discussed.**
>
> **A2.** It's not scientifically appropriate for studying data selection using checkpoint model. First, using an existing checkpoint makes it difficult to rigorously measure the effect of data selection, since the pretrained model may have already been trained on overlapping datasets, leading to data leakage issues [1] or dataset comtamination [2]. Second, large checkpoints are often too computationally expensive for our available resources. Although our current experiments are limited by hardware, the BLISS framework itself is scalable. Third, Many commercial large-scale pretrained models explicitly prohibit users from using their outputs to train other models, which may lead to severe legal and contractual consequences. For instance, OpenAI’s Terms of Use [3] disallow using any model output (Output) to develop competing models. Because of these restrictions, our “pretraining from-scratch” setting is sill useful: by pretraining our own models and using our own proxy-based data selection, we avoid these legal constraints entirely.
> We also note that BLISS naturally extends to continual pretraining, and we will update the revised version to clarify this connection.
>
> [1] Zeyuan Allen-Zhu and Yuanzhi Li. Physics of language models: Part 3.1, knowledge storage and extraction. arXiv preprint arXiv:2309.14316, 2023.
>
> [2] Yonatan Oren, Nicole Meister, Niladri Chatterji, Faisal Ladhak, and Tatsunori B Hashimoto. Proving test set contamination in black box language models. arXiv preprint arXiv:2310.17623, 2023.
>
> [3] OpenAI. OpenAI Terms of Service, 2024. URL https://openai.com/terms. Accessed: Jan 30, 2025.
>
> **Q3. Another limitation of this study is that the authors do not fully motivate the problem setting. While cleaning noisy data and removing near-duplicate examples are useful for LLM pretraining, I am not fully convinced why you have to select a subset of “quality data” from the full dataset. For example, if we do a coarse-grained filtering of the original dataset to leave around $70\\%-80\\%$ data and use all of them to do LLM pretraining, will the model performance better.**
>
> **A3.** A key motivation of our work is the realistic scenario in which pretraining compute is limited, making it infeasible to train on the full corpus. Under such a constraint, the central question becomes how to choose the most beneficial subset of data (e.g., $20\\%$) to maximize downstream performance. BLISS is designed specifically for this setting.
>
> We use the C4 dataset (Colossal Clean Crawled Corpus) for pretraining, which is derived from a single snapshot of Common Crawl and then cleaned via text filtering and de-duplication to remove spam, non-English text, boilerplate, and low-quality content. The filtering pipeline is documented by[6] and C4’s official dataset catalog at https://www.tensorflow.org/datasets/catalog/c4?utm_source=chatgpt.com#c4en_default_config.
>
> [6] Jesse Dodge, Maarten Sap, Ana Marasovi´c, William Agnew, Gabriel Ilharco, Dirk Groeneveld, Margaret Mitchell, and Matt Gardner. Documenting large webtext corpora: A case study on the colossal clean crawled corpus. arXiv preprint arXiv:2104.08758, 2021.
>
> **Q4. Can you try combining the validation tasks (e.g. take an average of their performances) for the purposes of targeting?**
>
> **A4.** We would like to clarify whether the reviewer is suggesting combining all validation tasks mentioned in the paper into a single aggregated validation set and using their averaged performance as the upper-level objective.
> If so, the resulting upper-level objective in formula (1) would become: $f(\theta\_p^{\star}(\theta\_s)) = \frac{1}{m}\sum\_{j=1}^{m}  \mathbb{E}\_{\zeta \sim \mathcal{D}^{j}\_{\mathrm{val}}}\ell(\theta\_p^{\star}(\theta\_s);\zeta)$,
> where $\mathcal{D}^{j}\_{\mathrm{val}}$ denotes the validation set for task $j$. The lower-level objective keep the same.
> Although this formulation is straightforward, each upper-level update would require evaluating the proxy model on all validation tasks, which substantially increases the computational cost per hypergradient step.
>
> We are currently conducting additional experiments following this setup and will update the revised manuscript once the results are ready.

---

> ### Author Response · Authors · 2025-11-27
>
> **Q5. Looking at the MATES paper, it seems you report the same exact numbers for their method. Are you using the exact same training implementation as they did? (Mentioned in NIPS)**
>
> **A5.** Yes. Our experimental settings (410M and 1B settings) are fully aligned with MATES, including the target pretraining model, pretraining data (C4), and compute budget (25B training tokens). In addition, for downstream evaluation, we directly use the official pretrained checkpoints provided by the MATES authors (linked at https://huggingface.co/yuzc19/pythia-410m-mates/tree/main and https://huggingface.co/yuzc19/pythia-1b-mates/tree/main), and the resulting downstream accuracies are identical to the numbers reported in their paper.
>
> **Q6. An assumption for the BLISS approach is that you want to model influence based on training to convergence. However, in practice, most LLMs are pre-trained based upon a finite compute budget rather than true “convergence” (due to the sheer quantities of data involved). Perhaps the authors could comment on this potential mismatch?**
>
> **A6.** We agree with the reviewer. Indeed, full convergence of a large LLM is unattainable with a limited compute budget in practice. Our method cannot assume convergence of the target LLM; instead, we approximate it by the multi-step training dynamics within each 3,000-step round (shown in Figure 6 and 7).  We have revised the claim of "trained to convergence" to "trained for a long time (i.e., multiple steps of gradient-based updates)" in line 70.

---

### Official Review · Reviewer_5Px2 · 2025-10-31

**Soundness:** 3
**Presentation:** 3
**Contribution:** 3
**Rating:** 6
**Confidence:** 3

**Summary:**

This paper introduces BLISS (Bilevel Influence Scoring for Data Selection), a lightweight approach for data selection consisting of two key components: a proxy model and a score model. The proxy model approximates the behavior of a large language model (LLM), while the score model assigns an influence score to each data sample. Experimental results indicate that the proposed method achieves superior performance across multiple downstream tasks.

**Strengths:**

1. The paper is well-written, clearly organized, and easy to follow.
2. The proposed bilevel influence scoring framework is conceptually novel and interesting.
3. The method is straightforward and intuitive—evaluating each data sample individually before selection.
4. Experimental results convincingly demonstrate the method’s effectiveness across several tasks.

**Weaknesses:**

1. The paper lacks a comparison with the state-of-the-art method [1], which also estimates the influence of individual data samples on model performance. Unlike BLISS, that approach computes influence scores directly on the original LLM using the method from [2], without relying on a proxy model.
2. The differences between BLISS and [1] are not sufficiently discussed, leaving unclear what advantages the bilevel formulation provides.
3. The motivation for training a proxy model instead of leveraging the original LLM is not well justified.
4. Experiments are conducted only on the C4 dataset, raising concerns about generalizability to other pretraining corpora.
5. The paper lacks an ablation study examining the effect of the number of training rounds or other key hyperparameters.

## Minor Issues
1. Lines 161–163: The functions F(-) and G(-) are mentioned but not introduced.



[1] Pan, Yanzhou, et al. "ALinFiK: Learning to Approximate Linearized Future Influence Kernel for Scalable Third-Parity LLM Data Valuation." Proceedings of the 2025 Conference of the Nations of the Americas Chapter of the Association for Computational Linguistics: Human Language Technologies (Volume 1: Long Papers). 2025.

[2] Lin, Huawei, et al. "Token-wise Influential Training Data Retrieval for Large Language Models." Proceedings of the 62nd Annual Meeting of the Association for Computational Linguistics (Volume 1: Long Papers). 2024.

**Questions:**

N/A

---

> ### Author Response · Authors · 2025-11-27
>
> **Q1. The paper lacks a comparison with the state-of-the-art method [1], which also estimates the influence of individual data samples on model performance. Unlike BLISS, that approach computes influence scores directly on the original LLM using the method from [2], without relying on a proxy model. The differences between BLISS and [1] are not sufficiently discussed, leaving unclear what advantages the bilevel formulation provides.**
>
> **A1.** ALinFiK [1] builds on RapidIn [2], which computes (or approximates) **per-sample influence** directly on **the full target LLM** via training/test gradient inner products, followed by regression to fit an influence model.
> Even with gradient compression, these steps must be performed on the large LLM, making the approach **computationally intensive**.
>
> In contrast, BLISS never computes influence on the target LLM. Instead we learn sample importance via a **lightweight proxy model-based bilevel optimization**, and the lightweight proxy model makes hypergradient (need to compute hessian-inverse product) computation tractable. A token-level KL term with the frozen target LLM ensures the proxy reflects the target model’s data preference without requiring any gradients from the large model.
>
> In summary, ALinFiK computes influence on the target LLM itself, whereas BLISS performs influence learning through a lightweight proxy model, making bilevel **optimization efficient** while remaining **aligned** with the target model. We have cited these papers in related work in the revised version.
>
> [1] Yanzhou Pan, Huawei Lin, Yide Ran, Jiamin Chen, Xiaodong Yu, Weijie Zhao, Denghui Zhang, and Zhaozhuo Xu. Alinfik: Learning to approximate linearized future influence kernel for scalable third-party llm data valuation. arXiv preprint arXiv:2503.01052, 2025.
>
> [2] Huawei Lin, Jikai Long, Zhaozhuo Xu, and Weijie Zhao. Token-wise influential training data re-trieval for large language models. arXiv preprint arXiv:2405.11724, 2024.
>
> **Q2. The motivation for training a proxy model instead of leveraging the original LLM is not well justified.**
>
> **A2.** The proxy model is necessary because bilevel optimization requires computing hypergradients, which involve Hessian-vector products and Jacobian-vector products in Eq. (3). Executing these second-order operations directly on the full-size target LLM would exceed GPU memory, making the bilevel updates infeasible.
> To address this, we introduce a lightweight proxy model on which second-order computations are memory-efficient, while the large target LLM is used only for forward KL alignment to ensure that the proxy keeps the target model’s data preference.
> This proxy model makes bilevel optimization efficient while keeping the same data preference with the target LLM.
>
> **Q3. Experiments are conducted only on the C4 dataset, raising concerns about generalizability to other pretraining corpora.**
>
> **A3.** We have conducted a new experiment of domain reweighting on Slimpajama-6B dataset [3], which contains $d=7$  domains, including arixv, book, C4, CC, Github, Stackexchange, and Wikipedia. We extended the bilevel data selection framework to the domain reweighting: the lower level objective is to optimize a lightweight proxy model (LLaMA-134M) given the domain weights, and the upper level tries to find the optimal domain weights $\alpha \in \mathbb{R}^d$ based on the updated proxy model from the lower level.
>
> We perform bilevel optimization for 1,000 steps to learn the domain weights, where $10\\%$ of the original training set is held out as the validation set for the upper-level objective, and the remaining $90\\%$ is used as the lower-level training set.
>
> Finally, we evaluate the resulting pretrained LLMs on the test set, and the results are shown in Figure 9 in Appendix J. The test loss curves show that data selection from our method outperforms random selection, achieving a lower test loss of 2.88 compared to 2.93 with random selection.
>
> [3]  DKYoon. Slimpajama-6b. HuggingFace Hub, 2023. https://huggingface.co/datasets/DKYoon/SlimPajama-6B.

---

> ### Author Response · Authors · 2025-11-27
>
> **Q4. The paper lacks an ablation study examining the effect of the number of training rounds or other key hyperparameters.**
>
> **A4.** The training round is determined by the partition of training data, where each training round performs data selection on one specific data partition. We follow exactly the same data partitioning protocol as MATES [4] using the C4 configuration provided by DsDM [5] at the link https://huggingface.co/datasets/loganengstrom/dsdm-candidate-c4.
>
> We did the ablation study to investigate the steps of bilevel optimization. As shown in Figure 8 Appendix I, both the 3k-step and 5k-step settings converge to nearly the same training loss. This indicates that 3k steps are sufficient for the proxy model, as increasing the steps to 5k does not yield additional improvements. So we fix the training steps of bilevel optimization to 3k steps in main experiments.
>
> [4] Zichun Yu, Spandan Das, and Chenyan Xiong. Mates: Model-aware data selection for efficient pretraining with data influence models. arXiv preprint arXiv:2406.06046, 2024.
>
> [5] Logan Engstrom, Axel Feldmann, and Aleksander Madry. Dsdm: Model-aware dataset selectiot with datamodels. arXiv preprint arXiv:2401.12926, 2024.
>
> **Q5. Lines 161–163: The functions F(-) and G(-) are mentioned but not introduced.**
>
> **A5.** We have introduced them in revised version in line 162-163.

---

### Official Review · Reviewer_VCPD · 2025-11-01

**Soundness:** 3
**Presentation:** 3
**Contribution:** 2
**Rating:** 4
**Confidence:** 3

**Summary:**

The paper proposes BLISS, a novel bilevel optimization framework for data selection in large language model (LLM) pretraining. Unlike prior works that rely on external pretrained models (e.g., GPT-3.5) to assess data quality, BLISS operates entirely from scratch, using two lightweight models—a proxy model and a score model—to estimate the long-term influence of data samples. The proxy model mimics the target LLM’s behavior via KL divergence, while the score model learns to assign sample importance weights that optimize validation performance when the proxy model is trained to convergence. Experiments on Pythia (410M, 1B, 2.8B) and LLaMA-0.5B models show that BLISS outperforms baselines such as MATES, DSIR, and SemDeDup, achieving up to 1.7× faster convergence and 1.4% average accuracy gains on multiple downstream tasks, with reduced computational cost.

**Strengths:**

1. The use of a bilevel influence scoring mechanism for LLM data selection is innovative and theoretically grounded. It elegantly combines influence estimation with bilevel optimization.
2. BLISS avoids reliance on pretrained LLMs for scoring, addressing a key limitation in existing methods like QuRating and MATES.
3. Unlike single-step influence approximations, the framework explicitly models how data affects the model trained to convergence.
4. The authors present experiments across multiple model sizes (410M–2.8B) and architectures (Pythia, LLaMA), with consistent gains across nine downstream benchmarks.
5. Despite additional components (proxy and score models), BLISS achieves comparable or better performance with lower FLOPs.

**Weaknesses:**

1. Although bilevel optimization is central, the paper lacks a deeper convergence or approximation analysis for the surrogate models’ fidelity to full-scale LLMs.
2. While KL divergence is used for alignment, it remains unclear how well the proxy truly represents the LLM across domains. Quantitative metrics of proxy fidelity would strengthen the claims.
3. Training additional models (even small ones) adds complexity. The paper should analyze memory and runtime scalability for billion-scale pretraining setups.
4. The choice of validation data (e.g., LAMBADA) could bias data selection toward specific linguistic patterns. An analysis of robustness to different validation datasets would be useful.
5. The learned scores are treated as black boxes; understanding what kinds of data are preferred (e.g., factual, reasoning-heavy, stylistic) would improve interpretability.
6. Some ablation results (e.g., single-level vs bilevel) show marginal gains; the statistical significance of these improvements should be better substantiated.

**Questions:**

1. Add a theoretical or empirical analysis of proxy model representativeness.
2. Report runtime and memory scaling for the bilevel loop.
3. Discuss potential extensions to multimodal or instruction-tuning datasets, as mentioned in the conclusion.
4. Improve clarity in algorithmic descriptions—especially the stochastic hypergradient update (Eq. 5) which could benefit from pseudocode-level explanations.

---

> ### Author Response · Authors · 2025-11-27
>
> **Q1. Although bilevel optimization is central, the paper lacks a deeper convergence or approximation analysis for the surrogate models’ fidelity to full-scale LLMs.**
>
> **A1.** To verify the fidelity of proxy models to full-scale LLMs, we conduct a domain-reweighting experiment on SlimPajama-6B dataset [1], which contains $d=7$ domains: ArXiv, Books, C4, CommonCrawl, GitHub, StackExchange, and Wikipedia.
> The objective is to learn optimal domain weights $\alpha \in \mathbb{R}^d$ such that a model trained on data sampled according to the weights achieves the best downstream performance, i.e., the sample weight $P_i = \frac{\alpha_i}{\sum_{j=1}^{d} \alpha_j}$ in formula (1), where $i$ is the domain index.
>
> We compare two settings:
> - **Case 1 (with proxy model):**
>     The lower level optimizes a lightweight proxy model (LLaMA-134M) with output alignment to the target LLM (LLaMA-300M), and the upper level learns the domain weights $\tilde{\alpha}$.
>
> - **Case 2 (without proxy model):**
>     The lower level directly optimizes the target LLM (LLaMA-300M), and the upper level learns the domain weights $\alpha$.
>
> We perform bilevel optimization for 1,000 steps in both cases to learn the domain weights, where $10\\%$ of the original training set is held out as the validation set for the upper-level objective, and the remaining $90\\%$ is used as the lower-level training set.
> After obtaining $\tilde{\alpha}$ and $\alpha$, we train two final LLaMA-300M models on data sampled according to each set of weights.
> Figure 10 in Appendix J presents the learning curves of domain weights for both cases.
> We observe that the trajectories of $\tilde{\alpha}$ and $\alpha$ are highly similar across most domains (e.g., in the domain of wikipedia, Book, Stackexchange), demonstrating that the proxy model maintains high fidelity to the full-scale LLM in data selection.
>
> Finally, we evaluate the resulting pretrained LLMs on the test set, and the results are shown in Figure 9 in Appendix J. The test loss curves show that data selection based on the proxy model maintains high fidelity to the full-scale LLM, while significantly outperforming random selection.
>
> [1] DKYoon. Slimpajama-6b. HuggingFace Hub, 2023. https://huggingface.co/datasets/DKYoon/SlimPajama-6B.
>
>
> **Q2. While KL divergence is used for alignment, it remains unclear how well the proxy truly represents the LLM across domains. Quantitative metrics of proxy fidelity would strengthen the claims.**
>
> **A2.** Following the domain reweighting experiment, we compare the domain weights obtained using the proxy model ($\tilde{\alpha}$) with those obtained by directly optimizing the target LLM ($\alpha$). The resulting distributions are shown in Table 1, where the domain weights for random selection correspond to the uniform distribution $P$.
> The KL divergence between domain weights from the proxy-based and target-based is $D_{\mathrm{KL}}(\tilde{\alpha} \\| \alpha)=0.0098$,
> whereas the KL divergence between random selection and the target-based is $D_{\mathrm{KL}}(P \\|\alpha)=0.0424$.
>
> These results indicate that the domain weights learned via the proxy model exhibit high fidelity to those obtained directly from the full target LLM.

---

> > ### Author Response · Authors · 2025-11-27
> >
> > **Q6. Discuss potential extensions to multimodal or instruction-tuning datasets, as mentioned in the conclusion.**
> >
> > **A6.** Our bilevel data selection framework naturally extends to both multimodal pretraining and instruction-tuning cases. The key idea of BLISS is to select subset of training data that can maximize the validation (surrogate of the downstream task) performance, is model-agnostic and only requires (1) a proxy/score model, (2) a target model signal (e.g., KL or other matching loss), and (3) a validation objective. This framework generalizes as follows:
> >
> > - Multimodal: The lower-level loss becomes a cross-modal contrastive objective [2, 3], with KL divergence term remains unchanged. The proxy model will become a smaller CLIP-style image–text encoder, playing as a surrogate of the full scale of the target model.
> > - Instruction tuning: The lower-level becomes the SFT loss on instruction-response pairs, and validation uses held-out instruction benchmarks. Some Parameter-Efficient Fine-Tuning techniques like low rank adaptation (LORA) [4] can be applied to the proxy model, which will greatly reduce the memory  overhead for BLISS.
> >
> > [2] Chao Jia, Yinfei Yang, Ye Xia, Yi-Ting Chen, Zarana Parekh, Hieu Pham, Quoc Le, Yun-Hsuan Sung, Zhen Li, and Tom Duerig. Scaling up visual and vision-language representation learning with noisy text supervision. In International conference on machine learning, pp. 4904–4916. PMLR, 2021
> >
> > [3] Alec Radford, Jong Wook Kim, Chris Hallacy, Aditya Ramesh, Gabriel Goh, Sandhini Agarwal, Girish Sastry, Amanda Askell, Pamela Mishkin, Jack Clark, et al. Learning transferable visual models from natural language supervision. In International conference on machine learning, pp. 8748–8763. PmLR, 2021.
> >
> > [4] Edward J Hu, Yelong Shen, Phillip Wallis, Zeyuan Allen-Zhu, Yuanzhi Li, Shean Wang, Lu Wang, and Weizhu Chen. Lora: Low-rank adaptation of large language models. arXiv preprint arXiv:2106.09685, 2021.
> >
> > **Q7. Improve clarity in algorithmic descriptions—especially the stochastic hypergradient update (Eq. 5) which could benefit from pseudocode-level explanations.**
> >
> > **A7.** Thanks for your contructive suggestion, and we have added clarifying annotations to better explain the stochastic hypergradient update and improved the description of Algorithm 1 accordingly.

---

> ### Author Response · Authors · 2025-11-27
>
> **Q3. Training additional models (even small ones) adds complexity. The paper should analyze memory and runtime scalability for billion-scale pretraining setups.**
>
> **A3.**
> To address the reviewer’s concern on scalability, we measured the memory and runtime of the data selection stage for both BLISS and MATES under different target (or pretraining) model sizes (for short, T: target). The results are shown in Table 1. We have two observations:
>
> - BLISS scales well with larger target models. Note that the target model is not an optimization variable for the bilevel optimization and it is only  used  for calculating the KL divergence. Therefore, it does not affect the scalability of bilevel optimization. When increasing the target from 410M to 1B, BLISS’s memory and runtime grow moderately ($49.46\rightarrow 74.51$ GB; $5.03\rightarrow 11.82$ hours), as expected.
> - BLISS is significantly faster than MATES. MATES incurs high cost because each round requires oracle data collection. For every example, MATES performs a one-step gradient update on the target model and evaluates the validation loss change to compute influence scores. This per-example simulation dominates runtime. In contrast, BLISS avoids all per-example oracle evaluations in MATES and performs bilevel optimization directly on the proxy/score model, leading to $2-3 \times$ faster data selection.
>
> These results confirm that BLISS introduces modest overhead and scales well to billion-parameter settings. We have included the comparison of running time and memory in Appendix I.
>
> **Table1: Comparison of runtime and memory in the data selection stage.**
> | Setting             | Memory peak (GB) | Total data selection time (hours) |
> |---------------------|------------------|------------------------------------|
> | MATES: T: 410M      | 36.36            | 18.0                               |
> | MATES: T: 1B        | 63.52            | 30.32                              |
> | BLISS: T: 410M      | 49.46            | 5.03                               |
> | BLISS: T: 1B        | 74.51            | 11.82                              |
>
> **Q4. The choice of validation data (e.g., LAMBADA) could bias data selection toward specific linguistic patterns. An analysis of robustness to different validation datasets would be useful.**
>
> **A4.** We have already conducted this study in Appendix C.4, where we explicitly tested BLISS using four distinct validation datasets including SQuAD, ARC-E, LAMBADA, and PIQA, and compared their resulting downstream performance (Figure 5 in Appendix C.4).
>
> Across all four choices, BLISS consistently outperforms random selection on all downstream tasks, demonstrating that our method is robust to the choice of validation dataset. While LAMBADA yields the highest average accuracy (likely due to its broad domain coverage), the overall trend remains stable: the selected data distributions differ slightly, but downstream performance remains strong and comparable across validation sets.
>
> We believe this experiment addresses the reviewer’s concern: BLISS does not rely on any particular linguistic pattern of LAMBADA, and the method remains effective even when validation data vary in style and domain.
>
> **Q5. The learned scores are treated as black boxes; understanding what kinds of data are preferred (e.g., factual, reasoning-
> heavy, stylistic) would improve interpretability.**
>
> **A5.** In the domain reweighting experiment presented in Appendix J, the learned weights based on proxy model for Github, ArXiv, StackExchange, and Books are relatively high (approximately 0.174, 0.157, 0.167, and 0.148, respectively), while the weight for CommonCrawl is lower (around 0.123). This result arises because the validation set is a $10\\%$ split from the original training data and therefore shares the same domain distribution. In the training/validation set, Github, ArXiv, StackExchange, and Books constitute only $4.2\\%, 3.4\\%, 2.8\\%$, and $3.7\\%$ of the samples, whereas CommonCrawl accounts for $54.1\\%$ [1].
>
> Since the model can encounter a large amount of CommonCrawl data, learning this domain is relatively easy. Moreover, CommonCrawl primarily contains broad, general-purpose knowledge, whereas domains such as Github, ArXiv, StackExchange, and Books provide more specialized information that is underrepresented in the raw distribution. As a result, the bilevel optimization assigns a lower weight to CommonCrawl and higher weights to the domain-specific corpora.
>
> [1] DKYoon. Slimpajama-6b. HuggingFace Hub, 2023. https://huggingface.co/datasets/DKYoon/SlimPajama-6B.

---

### Meta-Review · Area_Chair_bMuf · 2026-01-13

**Summary:**

This paper proposes BLISS, a lightweight bilevel influence scoring approach for data selection in LLM pretraining. The method uses (i) a proxy model to make bilevel hypergradient computation tractable, (ii) a score model to assign example weights, and (iii) a KL alignment signal to keep the proxy consistent with the target LLM. The authors report experiments on C4 with Pythia (410M/1B/2.8B) and a smaller LLaMA setting, claiming faster convergence to a target performance level and improved downstream accuracy relative to several baselines (e.g., MATES, DSIR, SemDeDup). The rebuttal adds additional analyses on proxy fidelity (domain reweighting on SlimPajama), runtime/memory comparisons, and clarifies aspects of the framework.

**Reviewer Concerns:**

Addressed by the rebuttal: The authors clarified the motivation for using a proxy model, added empirical evidence of proxy fidelity via domain reweighting experiments, provided runtime and memory comparisons against MATES, expanded robustness analysis across different validation datasets, and clarified terminology around “trained to convergence.” They also addressed missing ablations, added discussion on extensions, and corrected clarity issues in the algorithmic description.

Still outstanding: Core concerns remain regarding overstated claims (“from scratch,” “long-term impact”), limited theoretical grounding specific to this setting, proxy representativeness across scales and architectures, modest gains relative to variance, incomplete comparisons with recent data selection methods, and lingering questions about the realism and generality of the pretraining/validation-set setup.

**Reviewer Scores:**

In my opinion, the reviewer scores would not have changed since many of the concerns still remain. For example, the generalization and comparison concerns, significance, pretraining realism, and validation bias, and the objections on claims, theory, and comparisons - these concerns would likely persist.

---

### Decision · Program_Chairs · 2026-01-26

Reject